# Spatial engineering of single-atom Fe adjacent to Cu-assisted nanozymes for biomimetic O$_2$ activation

Ying Wang [1,8], Vinod K. Paidi [2,8], Weizhen Wang [3], Yong Wang [1], Guangri Jia [4], Tingyu Yan [5], Xiaoqiang Cui [4], Songhua Cai [3] ✉, Jingxiang Zhao [5] ✉, Kug-Seung Lee [6] ✉, Lawrence Yoon Suk Lee [1,7] ✉ & Kwok-Yin Wong [1] ✉

The precise design of single-atom nanozymes (SAzymes) and understanding of their biocatalytic mechanisms hold great promise for developing ideal bio-enzyme substitutes. While considerable efforts have been directed towards mimicking partial bio-inspired structures, the integration of heterogeneous SAzymes configurations and homogeneous enzyme-like mechanism remains an enormous challenge. Here, we show a spatial engineering strategy to fabricate dual-sites SAzymes with atomic Fe active center and adjacent Cu sites. Compared to planar Fe–Cu dual-atomic sites, vertically stacked Fe–Cu geometry in FePc@2D-Cu–N–C possesses highly optimized scaffolds, favorable substrate affinity, and fast electron transfer. These characteristics of FePc@2D-Cu–N–C SAzyme induces biomimetic O$_2$ activation through homogenous enzymatic pathway, resembling functional and mechanistic similarity to natural cytochrome c oxidase. Furthermore, it presents an appealing alternative of cytochrome P450 3A4 for drug metabolism and drug–drug interaction. These findings are expected to deepen the fundamental understanding of atomic-level design in next-generation bio-inspired nanozymes.

Single-atom nanozymes (SAzymes), characterized by well-defined configuration and maximum atom-utilization efficiency, hold substantial promise in bridging the gap between heterogenous nanomaterials and homogenous enzymes, opening up the way for developing next-generation nanozymes[1,2]. Previous research efforts have focused on enhancing the intrinsic enzyme-like activities and specificity of SAzymes at the atomic level[3–5]. Considerable strategies mainly concentrated on metal active centers by tuning central metal types, non-metal heteroatoms doping[6], and regulation of coordination numbers over single-atom sites[7–10]. In practice, however, the active centers in many natural metalloenzymes typically function with protein scaffolds containing polymetallic sites[11], prompting the exploration of constructing dual-atomic geometry in SAzymes. Recent advances have successfully produced intriguing diatomic catalysts (DACs) through various synthesis methods, including high-temperature pyrolysis[12], atomic layer deposition, and wet-chemistry approaches, thereby

[1]State Key Laboratory of Chemical Biology and Drug Discovery, Department of Applied Biology and Chemical Technology, The Hong Kong Polytechnic University, Hung Hom, Kowloon, Hong Kong SAR, China. [2]European Synchrotron Radiation Facility, 71 Avenue des Martyrs, Grenoble 38043 Cedex 9, France. [3]Department of Applied Physics, The Hong Kong Polytechnic University, Hung Hom, Kowloon, Hong Kong SAR, China. [4]State Key Laboratory of Automotive Simulation and Control, Department of Materials Science, Key Laboratory of Automobile Materials of MOE, Jilin University, Changchun 130012, China. [5]Key Laboratory of Photonic and Electronic Bandgap Materials of MOE, College of Chemistry and Chemical Engineering, Harbin Normal University, Harbin 150025, PR China. [6]Pohang Accelerator Laboratory (PAL), Pohang University of Science and Technology (POSTECH), Pohang 37673, Republic of Korea. [7]Research Institute for Smart Energy, The Hong Kong Polytechnic University, Hung Hom, Kowloon, Hong Kong SAR, China. [8]These authors contributed equally: Ying Wang, Vinod K. Paidi. ✉e-mail: songhua.cai@polyu.edu.hk; xjz_hmily@163.com; lks3006@postech.ac.kr; lawrence.ys.lee@polyu.edu.hk; kwok-yin.wong@polyu.edu.hk

breaking the linear relationship limitation in heterogenous catalysis[13]. Those studies have led to the discovery of diatomic iron nanozymes[14,15], Fe−Co dual-sites nanozymes[16], Fe−Cu hetero-binuclear nanozymes[17], and Mo/Zn dual SAzymes with multienzyme-mimetic biocatalysis[18]. Despite these notable strides, the development of SAzymes with dual-sites is still at an early stage, and additional progress is necessary to shrink the gap between heterogeneous and homogeneous biocatalysis.

Dioxygen (O₂) activation plays an integral role in oxidative phosphorylation and cellular respiration in biological systems[19]. Natural cytochrome c oxidases (CcO) are the terminal metalloenzymes for these reactions, which are composed of hemes (heme a and heme a₃) and copper centers (Cu_A and Cu_B)[20]. Moreover, a binuclear center of the cytochrome a₃ and Cu_B acts as the oxygen-binding site for the four-electron O₂ reduction to water ($O_2 + 4H^+ + 4e^- \rightarrow 2H_2O$) without any release of partially reduced oxygen species (PROS)[21,22]. The achievement of homogeneous enzymatic features relies on the synergistic integration of well-arranged components within a three-dimensional (3D) pocket, which includes central metal units, covalently bonded ligands in the first coordination shell, and appropriate binding sites through non-covalent interactions in the second shell coordination[20]. In this regard, the construction of desirable nanozymes for efficient O₂ activation demands fulfillment of two critical requirements: (1) The active sites should adopt 3D spatial Fe−Cu binuclear topologies, which are responsible for the activation and reduction of dioxygen to water[23]; (2) The corresponding microenvironment of SAzymes should be configured to mimic the efficient proton-coupled electron transfer (PCET) pathway in a homogeneous enzyme-like mechanism[24]. It is noteworthy that introducing axial ligand coordination as cofactors into single-atom configuration has shown promising results for engineering enzyme-like performances[25–27]. However, SAzymes designed using reported spatial regulation strategies have primarily focused on mimicking partial bio-inspired structures of natural enzymes[28–31], largely overlooking the multi-spatial dimensionality and homogeneous enzymatic pathway. In essence, it is both feasible and meaningful to explore a more versatile spatial engineering strategy that can concurrently integrate heterogeneous SAzyme configurations and homogeneous enzyme-like mechanisms, thereby unlocking their full O₂ activation capacity.

In this work, we have developed a comprehensive spatial engineering strategy to fabricate dual-site SAzymes. These SAzymes incorporate single-atom Fe active centers (Fe-N₄) and Cu atomic species (Cu–N₄) with distinct spatial configurations. Both experiments and theoretical results indicate the dual-site SAzyme featuring vertically stacked Fe-N₄ and Cu-N₄ geometry (FePc@2D-Cu–N-C) exhibits stronger electronic coupling and synergistic interaction compared with planar Fe–Cu pairs in two-dimensional (2D) architectures (2D-FeCu-N–C), thus enabling a similar electron transfer process to that observed in natural CcO. This spatial configuration of FePc@2D-Cu-N-C SAzyme leads to enhanced oxidase-like performance, surpassing the conventional 2D-FeCu-N-C and single-atomic Fe/Cu counterparts. Systematic kinetic investigations further unveil a highly favorable binding affinity to enzyme substrates and strong O₂ activation on the FePc@2D-Cu–N–C. Similar to natural CcO-like reaction pathway, the FePc@2D-Cu–N–C SAzyme facilitates four-electron O₂ reduction to H₂O with low production of toxic PROS, achieving homogenous biomimetic O₂ activation. More importantly, in a proof-of-concept application of drug metabolism, this as-developed FePc@2D-Cu-N-C SAzyme also demonstrates remarkable cytochrome P450 3A4 (CYP3A4)-like activities. As a potential alternative to CYP3A4, the proposed FePc@2D-Cu-N-C SAzyme can be employed to regulate the drug–drug interaction (DDI) and explore the corresponding mechanism of inhibitory behaviors.

## Results

### Preparation and morphological characterization

The synthetic procedures of Cu single atoms stabilized by N atoms into carbon nanosheets (2D-Cu-N-C) and iron phthalocyanine supported on 2D-Cu-N-C (FePc@2D-Cu-N-C) are illustrated in Fig. 1. A Cu-containing precursor is first prepared by facile salt-template confinement. The carbonization of the precursor, followed by acid washing to remove the template and impurities, yields large-area 2D-Cu-N-C

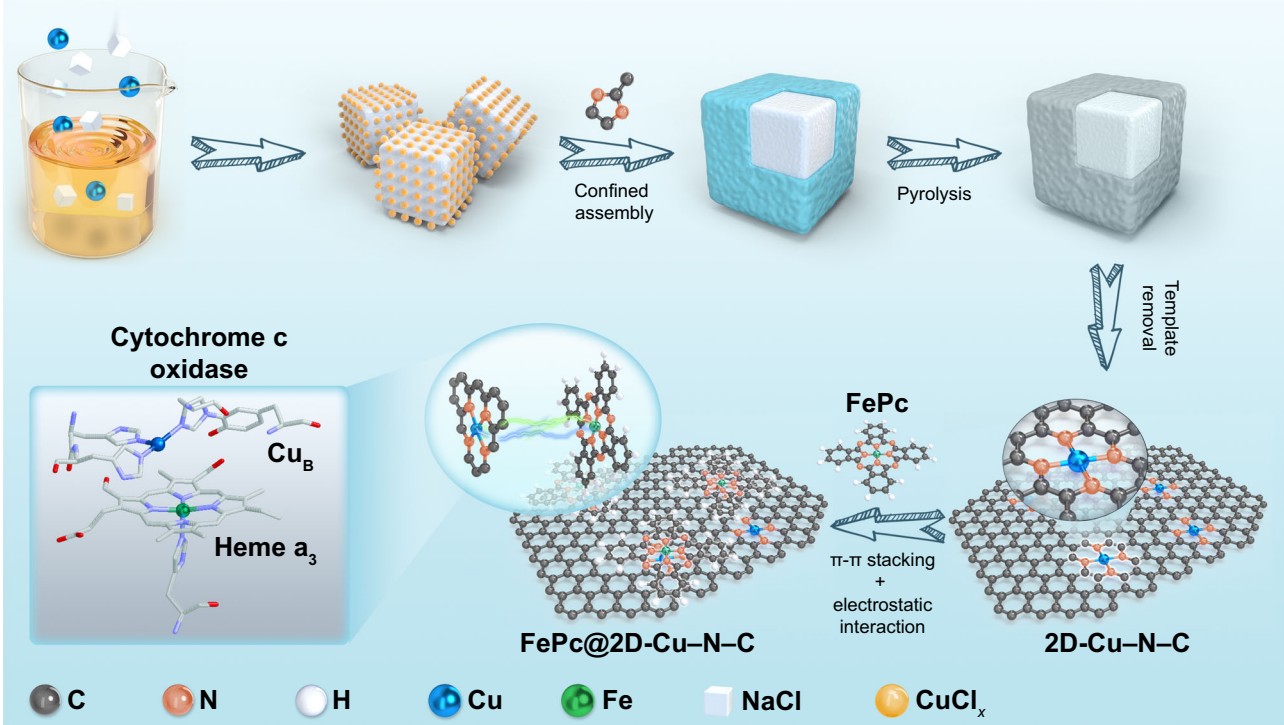

Confined assembly

Pyrolysis

Template removal

**Cytochrome c oxidase**

Cu_B

Heme a₃

FePc

π-π stacking + electrostatic interaction

**FePc@2D-Cu–N–C** **2D-Cu–N–C**

● C ● N ○ H ● Cu ● Fe ☐ NaCl ● CuCl_x

**Fig. 1 | A schematic diagram for spatial engineering of FePc@2D-Cu–N–C.** The schematic synthesis of 2D-Cu-N-C and FePc@2D-Cu-N-C SAzymes (PDB entry 1OCR).

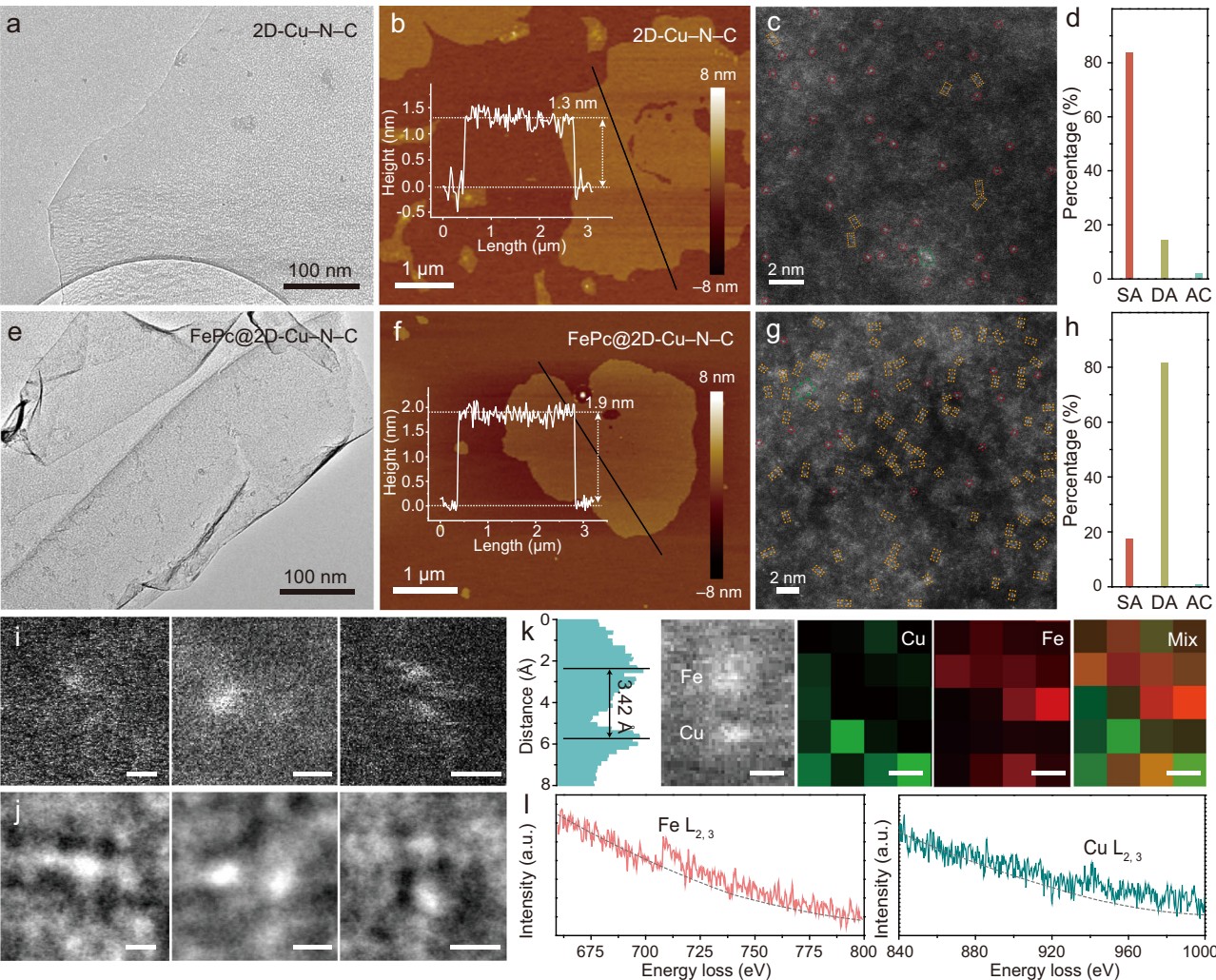

**Fig. 2 | Morphology of 2D-Cu–N–C and FePc@2D-Cu–N–C. a, e** TEM images, **b, f** AFM images, **c, g** AC-HAADF-STEM images of 2D-Cu–N–C and FePc@2D-Cu–N–C (*n* = 3 images from three independent samples). **d, h** Statistical study of the single-atom (SA, red circles), dual-atom (DA, orange rectangles), and atomic clusters (AC, green rectangles) in **c** and **g**. **i, j** The HAADF-STEM images (**i**) and iDPC-STEM images (**j**) recorded on an FePc@2D-Cu–N–C catalyst, scale bar: 2 Å (three times each experiment was repeated independently with similar results, and representative results are given). **k–l** Magnified HAADF-STEM image and intensity profile of Fe–Cu atom pairs in the FePc@2D-Cu–N–C, simultaneously acquired HAADF-STEM image with atomic-resolution EELS elemental mapping (Cu: green, Fe: red) (**k**) and the corresponding EELS spectra (**l**). 'a. u.' represents arbitrary units. Source data are provided as a Source Data file.

nanosheets (10 μm²). The 2D-Cu–N–C shows a smooth surface with ultrathin layer from as-obtained scanning electron microscopic (SEM) and transmission electron microscopic (TEM) images (Supplementary Fig. 1a, Fig. 2a). The predominant thickness of 2D-Cu–N–C is 1.3 nm by atomic force microscopy (AFM) (Fig. 2b). The diffraction peaks at 25° and 42° in the powder X-ray diffraction (XRD) pattern are attributed to graphitic carbon, respectively[32], and no additional peaks due to secondary phases are observed (Supplementary Fig. 1b). The TEM images further confirm that no clusters or nanoparticles are present, suggesting no metal agglomeration to crystalline metallic species in 2D-Cu–N–C (Supplementary Fig. 1c). Energy-dispersive X-ray spectroscopic (EDS) mapping verifies the uniform dispersion of C, N, and Cu species into the entire 2D-Cu–N–C matrix (Supplementary Fig. 2). Moreover, the aberration-corrected high-angle annular dark-field scanning transmission electron microscopy (HAADF-STEM) image displays ample bright spots distinguished from the carbon matrix of 2D-Cu–N–C (Fig. 2c)[33]. The statistical analysis reveals that over 80% of the identifiable bright spots representing single-atom motifs are isolated (5.83 wt%). This indicates the atomic-level distribution of Cu single atoms, which is attributed to their large atomic number (Fig. 2d

and Supplementary Table 1). For comparison, Fe single atoms are also synthesized using the same approach (denoted as 2D-Fe–N–C, Supplementary Figs. 3 and 4).

This salt-template-assisted strategy is further extended to prepare dual-metal single-atomic catalysts in which both Cu and Fe single atoms are more likely to co-exist in-planar layers of the ultrathin 2D-N–C nanosheet (denoted as 2D-FeCu–N–C). As depicted in Supplementary Fig. 5, the 2D sheet-like morphology is well maintained, and metal-based species are absent in this 2D-FeCu–N–C. Interestingly, a higher population of bright spots is observed from the HAADF-STEM image of 2D-FeCu–N–C compared with 2D-Cu–N–C (Supplementary Fig. 6a). The analysis reveals that >80% of the observed dots correspond to atomic pairs (yellow rectangles; defined as distance <5 Å), suggesting the formation of dual-atomic sites within the planar 2D-FeCu–N–C (Supplementary Fig. 6b)[34]. The EDS elemental mapping images evidence the homogeneous distribution of C, N, Cu, and Fe over the whole carbon matrix (Supplementary Fig. 6c).

To obtain the vertically stacked coordination geometry of Cu and Fe single atoms, another facile route was employed using iron

phthalocyanine (FePc) molecules with a planar tetracoordinate Fe-$N_4$ center[35]. The negatively charged FePc molecules were assembled onto the positively charged 2D-Cu-N-C through strong non-covalent π-π stacking and electrostatic interaction (FePc@2D-Cu-N-C, Supplementary Fig. 7a, b)[36,37]. The FePc@2D-Cu-N-C exhibits an ultrathin sheet-like morphology with a slightly rougher surface and higher thickness than pristine 2D-Cu-N-C (Fig. 2e, f, Supplementary Fig. 7c). The amount of adsorbed FePc is quantitatively optimized to ensure their homogeneous distribution on the 2D-Cu-N-C without agglomeration (Supplementary Figs. 8 and 9). The corresponding XRD patterns and Raman spectra display no additional peaks from the aggregated FePc other than two broad peaks from the carbon support (Supplementary Fig. 9b-c). Fourier-transform infrared (FT-IR) spectroscopy identifies the peaks at 1339.6, 1123.6, and 668.5 cm$^{-1}$ from FePc@2D-Cu-N-C, which represent the C=N and C-H vibrations of FePc moieties (Supplementary Fig. 9d). The HAADF-STEM image of FePc@2D-Cu-N-C displays densely populated bright dots counted with >80% paired bimetallic sites (Fig. 2g, h)[38]. The EDS elemental mapping images confirm that C, N, Fe, and Cu elements are homogeneous in the whole area of FePc@2D-Cu-N-C (Supplementary Fig. 10). It is worth noting that electron microscopic images provide a 2D projection along the incident beam direction[39]. The coaxial line of FePc and 2D-Cu-N-C layers may not align along the projection direction because of the overlapping and ripple of 2D-Cu-N-C substrates. Further considering the limitations of HAADF-STEM in accurately recognizing light elements in FePc@2D-Cu-N-C based on Z contrast (Z, atomic number), we employed another STEM imaging mode, low-dose integrated differential phase contrast-scanning transmission electron microscopy (iDPC-STEM)[40,41]. In Fig. 2i, j, the magnified iDPC-STEM images provide a much enhanced signal-to-noise ratio for the spatial characteristics of FePc@2D-Cu-N-C compared with the HAADF-STEM images. Detailed features of Fe-Cu bimetallic pairs in the vertically stacked FePc@2D-Cu-N-C are highly dependent on their orientations[42]. Additionally, HAADF-STEM with atomic-resolution electron energy-loss spectroscopy (EELS) elemental mapping was employed at a relatively low beam current[43]. The spatial configuration of Fe and Cu species in FePc@2D-Cu-N-C are highly correlated (Fig. 2k, l), which confirms the formation Fe-Cu single-atom pairs in FePc@2D-Cu-N-C, closely resembling the spatial configuration observed in natural CcO[44].

## Structural characterization

To determine the chemical composition and local environment, X-ray photoelectron spectroscopy (XPS) and X-ray absorption spectroscopy (XAS) analysis were proceeded. In C 1s and O 1s regions of XPS spectra, the characteristic peaks for the metal-based carbides and oxides are absent in all samples (Supplementary Fig. 11). The high-resolution N 1s spectrum of FePc presents two major types of N species, pyridinic N (398 eV) and pyrrolic N (399.4 eV, Fig. 3a). Interestingly, the primary N type in FePc@2D-Cu-N-C is pyridinic N, whereas the majority in 2D-FeCu-N-C is the graphitic N, which suggests the possibility of different local environments in two configurations (Supplementary Fig. 12)[45]. The Fourier transformed k3-weighted (FT) extended X-ray absorption fine structure (EXAFS) spectra at the Fe K-edge of 2D-FeCu-N-C and FePc@2D-Cu-N-C show a prominent peak at ~1.5 Å (not corrected for phase shift) due to the Fe-N coordination at the first shell (Fig. 3b). Similarly, the FT-EXAFS spectra of Cu K-edge in 2D-Cu-N-C, 2D-FeCu-N-C, and FePc@2D-Cu-N-C display a main peak at ~1.5 Å (not corrected for phase shift) due to the Cu-N coordination (Fig. 3c). All wavelet transform (WT) contour plots at the Fe/Cu K-edge of 2D-FeCu-N-C, and FePc@2D-Cu-N-C exhibit one major intensity maximum at 3.9 Å$^{-1}$ (4.2 Å$^{-1}$) assigned to the Fe-N (Cu-N) pair without Fe-Fe (Cu-Cu) signals (Supplementary Fig. 13)[46]. Both FT- and WT-EXAFS analyses prove that the Fe and Cu atoms are isolated, which agrees with HAADF-STEM results.

In order to unveil the precise coordination geometry, the EXAFS data were fit to density functional theory (DFT) optimized models[47,48]. As obtained in the Fe K-edge and Cu K-edge EXAFS fitting, the first coordination shell of both 2D-FeCu-N-C and FePc@2D-Cu-N-C is attributed to the Fe-$N_4$ and Cu-$N_4$ scattering path and local environment (Fig. 3d, e, Supplementary Figs. 14-17, and Supplementary Table 2 to Table 3). Additionally, a mixture of Fe-N, Fe-C, and Fe-Cu coordination paths are identified in the second coordination shell of FePc@2D-Cu-N-C. The interlayer spacing of 3.08 Å corresponds to the most stable stacking configuration between axial FePc and planar 2D-Cu-N-C (Supplementary Fig. 18). To further verify the configuration of FePc@2D-Cu-N-C, $^{57}$Fe Mössbauer spectroscopy is performed for characterizing spin polarization configuration of Fe species with ultrahigh sensitivity[49]. As depicted in Fig. 3f, the Mössbauer spectrum of FePc@2D-Cu-N-C can be well deconvoluted into three doublets according to the isomer shift (IS) and quadrupole splitting (QS) values (Supplementary Table 4). The doublet of D3 (green) is assigned to the FePc-like square-planar Fe(II)$N_4$ moiety[50]. The two prominent D2 (blue) and D1 (yellow) doublet peaks are assigned to the X-FeN$_4$ species with a penta-coordinated rhombic mono-pyramidal structure (X = N or axial ligands) and X-FeN$_4$-adsorbed $O_2$ molecule (X-FeN$_4$-$O_2$), respectively[51]. These results strongly suggest the vertical stack configuration of FePc@2D-Cu-N-C and strong interaction between FePc centers and 2D-Cu-N-C supports. In comparison, the EXAFS modeling results of 2D-FeCu-N-C reveal that the second coordination shell can be satisfactorily interpreted as the Fe-C contribution and Fe-Cu interatomic distances (~4.1 Å). All the features of the experimental Fe and Cu K-edges X-ray absorption near edge structure (XANES) spectra are reproduced by the XANES simulation using the DFT-optimized 2D-FeCu-N-C structures (Supplementary Fig. 19). More specifically because XANES spectroscopy is sensitive to the 3D arrangement of atoms around the absorbing atom, we compare the detailed XANES spectral features to better identify the atomic structure of the samples (Supplementary Fig. 20)[33]. This analysis concurrently reveals that the planar 2D-FeCu-N-C and axial FePc@2D-Cu-N-C are successfully synthesized with evenly atomic distributed Fe and Cu atoms.

Afterwards, the electronic structure of Fe and Cu atoms was further unveiled by the linear relationship between the XANES absorption threshold position and metal valence states. Notably, the valence state of Fe species in FePc@2D-Cu-N-C increases to higher energies than that of the FePc reference, which indicates the depletion of electrons of axial FePc molecules (Fig. 3g and Supplementary Fig. 21a)[45]. In agreement with the Fe K-edge spectra, a decreased oxidation state of Cu is observed from the FePc@2D-Cu-N-C compared with 2D-Cu-N-C (Fig. 3h and Supplementary Fig. 21b). In contrast, when the 2D-Cu-N-C supports are replaced with another 2D carbon-based material, carbon nitride ($C_3N_4$), the absorption edge position of FePc@$C_3N_4$ remains similar to the FePc reference (Supplementary Fig. 22). These results provide evidence that the incorporation of more electronegative Cu elements is believed to effectively accelerate the electron transfer and modulate the local environment of Fe centers. Such Cu-induced synergistic effects unambiguously promote the charge polarization and transfer from the axial Fe center to the single-atomic Cu sites in 2D-Cu-N-C supports in the FePc@2D-Cu-N-C configuration. These observations are similar to the natural cytochrome oxidase, where the electron transfer between cytochrome c (Cyt c) and the heme/Cu site is mediated by two redox cofactors (Cu$_B$ and heme a)[21,52].

## Investigations of the oxidase-like performance and kinetics analysis

The oxidase (OXD)-like activities of single- and dual-atomic nanozymes are systematically investigated using a colorimetric assay, in which 3,3′,5,5′-tetramethylbenzidine (TMB) is oxidized to a typical blue product ($_{ox}$TMB)[4]. The FePc@2D-Cu-N-C demonstrates markedly higher

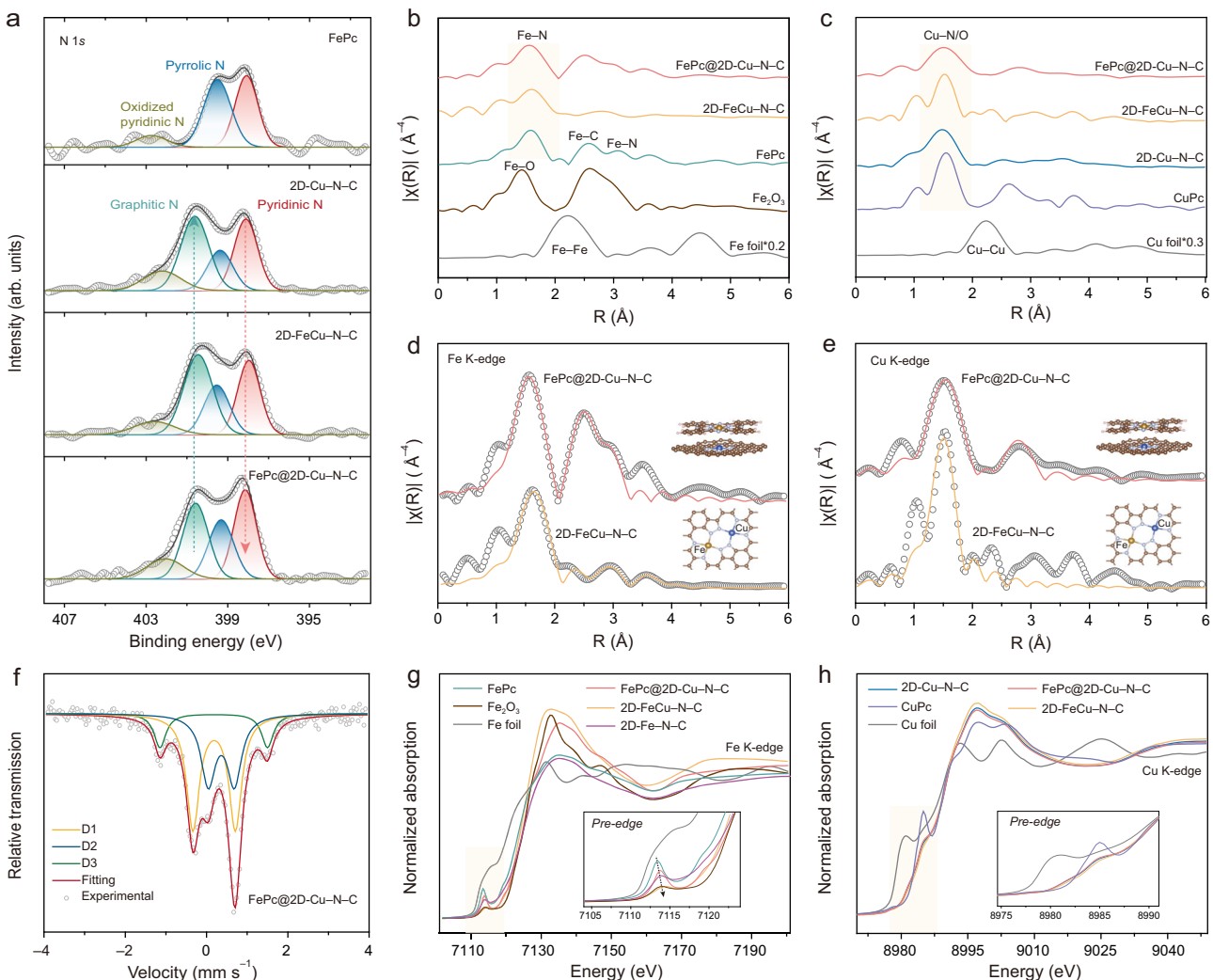

**Fig. 3 | Structural characterization of 2D-FeCu–N–C and FePc@2D-Cu–N–C.**
**a** XPS spectra of FePc, 2D-Cu–N–C, 2D-FeCu–N–C, and FePc@2D-Cu–N–C catalysts in the N 1s regions. **b, c** The Fourier transformed magnitude of the $k^3$-weighted EXAFS spectra of the Fe K-edge (**b**) and Cu K-edge (**c**) of FePc, 2D-Cu–N–C, 2D-FeCu–N–C, FePc@2D-Cu–N–C, and the corresponding reference samples. **d, e** Fe K-edge (**d**) and Cu K-edge (**e**) EXAFS experimental results (open circles) and theoretical fits (solid lines) of 2D-FeCu–N–C and FePc@2D-Cu–N–C in *R* space (Inset: the corresponding optimized configuration). **f** $^{57}$Fe Mössbauer spectrum of FePc@2D-Cu–N–C showing the contributions from three Fe species, D1–D3. **g, h** Normalized Fe K-edge (**g**) and Cu K-edge (**h**) XANES spectra of various catalysts with the corresponding reference compounds. Insets are the magnified corresponding pre-edge regions. Source data are provided as a Source Data file.

OXD-like performance as manifested by the highest characteristic absorbance at 652 nm compared with other counterparts (Fig. 4a). The intrinsic OXD-like activity of FePc@2D-Cu–N–C is also tested by evaluating the oxidation of two typical oxidase substrates, $O_2^-$, pH-, and temperature-dependent OXD-like behavior (Supplementary Fig. 23 to Fig. 24). The corresponding TEM image, XANES and EXAFS spectra further confirm the remarkable stability of FePc@2D-Cu–N–C nanozyme (Supplementary Figs. 25–26). Besides, the OXD-like activity was quantitatively measured by evaluating specific activity (SA) values (U mg$^{-1}$, Supplementary Fig. 27)[1]. The SA of axial FePc@2D-Cu–N–C is estimated ~2.42 and 1.34 times higher than that of parental 2D-Cu–N–C support and planar 2D-FeCu–N–C nanozyme, respectively (Fig. 4b). To elucidate the factors contributing to the enhanced OXD-like catalytic efficiency, a series of control analysis were performed to confirm the indispensability of each component in FePc@2D-Cu–N–C (Supplementary Figs. 28–30). In particular, a series of steady-state kinetic constants including the maximum reaction velocity ($V_{max}$), Michaelis–Menten constant ($K_m$), catalytic rate constant ($k_{cat}$), and catalytic efficiency ($k_{cat}/K_m$) were calculated and compared in Supplementary Fig. 31 and Supplementary Table 5[53]. FePc@2D-Cu–N–C delivers the lowest $K_m$ value (0.667 mM) among all nanozymes studied, denoting the highest affinity toward TMB substrate (Fig. 4c). Meanwhile, all the kinetic indicators of catalytic efficiency ($V_{max}$, $k_{cat}$, and $k_{cat}/K_m$) are the highest for FePc@2D-Cu–N–C, which indicates not only its improved binding affinity to the enzyme-substrate but also the facilitated activation of $O_2$.

## Experimental and theoretical analysis for the $O_2$ activation mechanism

In general, the natural CcO metalloenzyme can directly catalyze the four-electron $O_2$ reduction to $H_2O$ without releasing toxic PROS[24,54]. The electron transfer to the heme is one of the rate-limiting steps for the whole respiration reaction. Insufficient electron transfer would lead to the generation of PROS; superoxide ($O_2^{\cdot-}$) is produced via one-electron reduction, and hydroxyl radical (·OH) is generated via three-electron reduction[54]. To gain further insights into the intrinsic OXD-like behaviors of axial FePc@2D-Cu–N–C and in-planar 2D-FeCu–N–C, electron spin resonance (ESR) and the corresponding control ROS-trapping experiments were performed to identify the potential PROS generated[55]. The characteristic peaks of 5,5-dimethyl-1-pyrroline

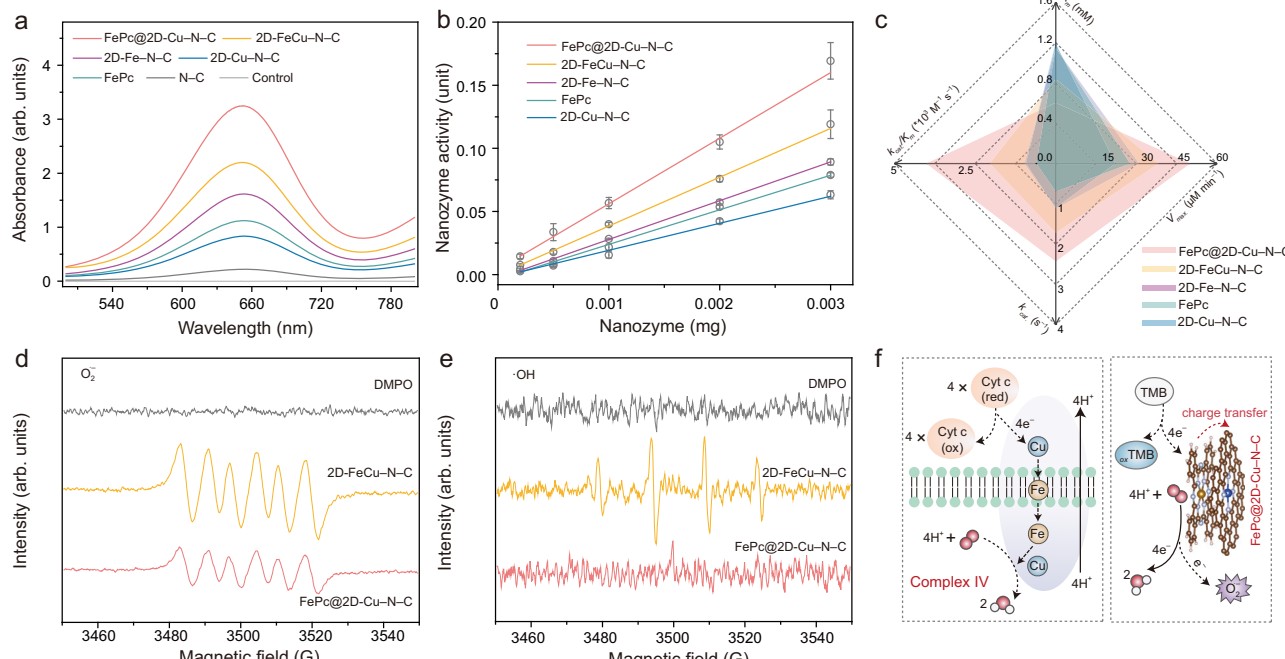

**Fig. 4 | Oxidase-like performance and kinetics analysis. a, b** Absorption spectra of $_{ox}$TMB (**a**) and specific activities (SA, U mg$^{-1}$ metal atoms) (**b**) of N–C, 2D-Cu–N–C, FePc, 2D-Fe–N–C, 2D-FeCu–N–C, and FePc@2D-Cu–N–C ($n = 3$ independent experiments and all data are presented as mean values ± SD). **c** A spidergram comparing the parameters of the Michaelis–Menten kinetics (Michaelis constant ($K_m$), maximum reaction rates ($V_{max}$), catalytic rate constant ($k_{cat}$), and catalytic efficiency ($k_{cat}/K_m$)). **d, e** ESR spectra of 2D-FeCu–N–C and FePc@2D-Cu–N–C for the detection of O$_2^{\cdot-}$ in the methanol system (**d**) and •OH in the HAc-NaAc buffer (pH = 4.0) system (**e**) trapped by DMPO. **f** Schematic diagram illustrating the OXD-like properties of FePc@2D-Cu–N–C and natural CcO. Source data are provided as a Source Data file.

N-oxide (DMPO)-O$_2^{\cdot-}$ are observed from both ESR spectra taken in the presence of FePc@2D-Cu–N–C and 2D-FeCu–N–C (Fig. 4d). The signal intensity is, however, weaker with FePc@2D-Cu–N–C than 2D-FeCu–N–C. Furthermore, 2D-FeCu–N–C displays quartet ESR signals with the relative signal intensity of 1:2:2:1, indicating the existence of DMPO-•OH in the OXD-like catalytic reaction system, whereas no such signal of •OH is captured in the case of FePc@2D-Cu–N–C (Fig. 4e). On the other hand, the addition of superoxide dismutase (SOD), which specifically catalyzes the disproportionation of O$_2^{\cdot-}$ species[56], decreases the OXD-like catalytic rate of 2D-FeCu–N–C more significantly than that of FePc@2D-Cu–N–C (Supplementary Fig. 32). This implies that fewer O$_2^{\cdot-}$ species are produced by the OXD-like enzymatic activities of FePc@2D-Cu–N–C. These observations suggest the possibilities of dissimilar OXD-like reaction pathways and mechanisms between the 2D-FeCu–N–C and FePc@2D-Cu–N–C configurations.

As depicted in Fig. 4f, the natural CcO can receive electrons from four Cyt c molecules and transfer them to one oxygen molecule and four protons, resulting in the production of two water molecules. The cytochrome a$_3$ and Cu$_B$ form a binuclear center that acts as the oxygen reduction site. From a kinetic perspective, nanozymes also have the ability to accept electrons from TMB substrates (acting as the electron donor) and undergo partial reduction in the initial step of the OXD-like reaction (denoted as reaction rate 1, $v_1$). Then, these electrons are delivered to oxygen molecules that are bound to the surface of nanozymes via various electron transfer pathways (denoted as reaction rate 2, $v_2$). The axial FePc molecule on FePc@2D-Cu–N–C nanozymes can serve as an effective mediator for electron transfer. Due to its negative charge, the axial FePc molecule tends to non-covalently couple with positively charged TMB by electrostatic interaction and π–π stacking[57]. The experimentally determined K$_m$ values further confirm that axial FePc@2D-Cu–N–C exhibits a higher affinity toward TMB and higher $v_1$ value compared to the in-planar 2D-FeCu–N–C. With FePc@2D-Cu–N–C, the four-electron O$_2$ reduction to H$_2$O is more likely to occur due to its high $v_2$ value, resulting in lower levels of toxic PROS. In contrast, the rate-limiting $v_1$ value on in-planar 2D-FeCu–N–C leads to insufficient reduction and slower $v_2$ rates. Consequently, this causes the production of multiple PROS and a slower oxygen reduction process (Supplementary Fig. 33). Another difference between FePc@2D-Cu–N–C and 2D-FeCu–N–C is their wettability. Remarkably, the FePc@2D-Cu–N–C displays hydrophobic properties with a high water contact angle (CA = 91.3°), whereas the 2D-FeCu–N–C shows relative hydrophilic property (CA = 62.8°, Supplementary Fig. 34). The natural CcO is protected by a membrane that can prevent the active site from hydrolytic autoxidation[54]. The axial FePc molecule on FePc@2D-Cu–N–C can serve as a hydrophobic blocking layer, thereby decreasing the formation of PROS.

We further employed DFT calculations to explore the OXD-like catalytic mechanisms of FePc@2D-Cu–N–C and 2D-FeCu–N–C SAzymes. For the overall OXD-like enzymatic reaction pathway, the electron transfer from substrates and the sufficient decomposition of O$_2$ are two major processes[4]. The electron transfer capacity was evaluated by the work function (designated as Φ) based on the calculated electrostatic potential (Fig. 5a and Supplementary Fig. 35). Clearly, FePc@2D-Cu–N–C displays the highest Φ value of 4.30 eV compared with 2D-FeCu–N–C and single-atom counterparts. In addition, ultraviolet photoemission spectroscopy (UPS) was engaged to illustrate the energy level differences (Fig. 5b). Based on the cutoff energies ($E_{cutoff}$) of FePc@2D-Cu–N–C (17.88 eV) and 2D-FeCu–N–C (18.01 eV), their work functions are determined as 3.34 and 3.21 eV, respectively, which can infer that the electron transfer from TMB to the metal sites in FePc@2D-Cu–N–C is more preferable[58]. The corresponding adsorption energies ($E_{ads}$) of TMB substrates further confirm that FePc@2D-Cu–N–C interacts with TMB more strongly than 2D-FeCu–N–C, which is in good accordance with the experimental $K_m$ values (Fig. 5c).

Generally, the redistribution of local electron density results in the rearrangement of d-orbital electron configuration, and thus enhances

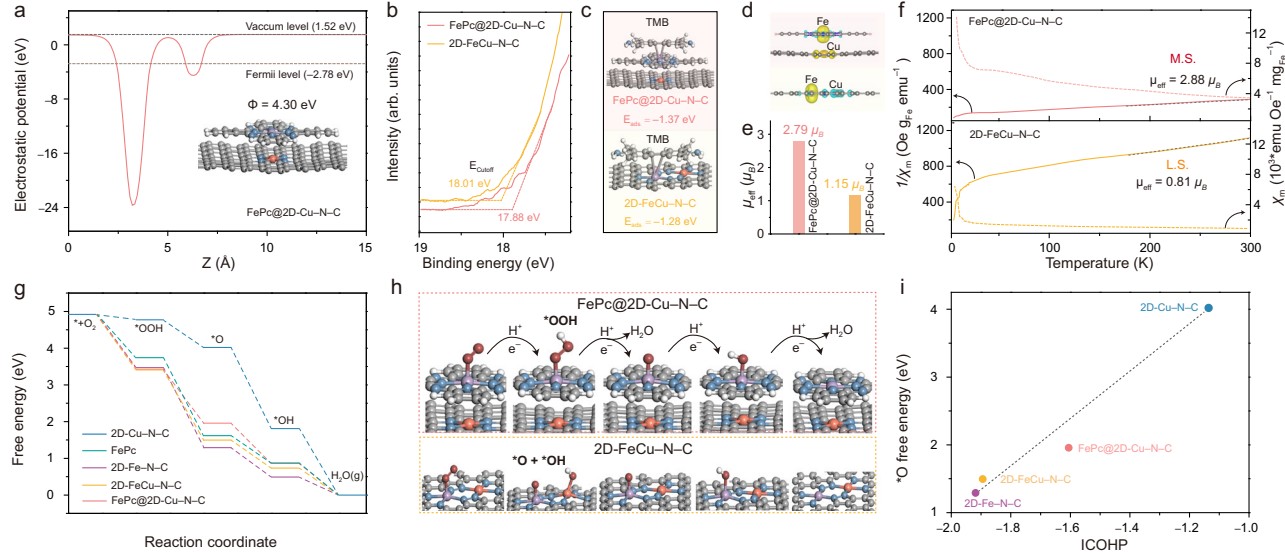

**Fig. 5 | Experimental and theoretical analysis for the O$_2$ activation mechanism of 2D-FeCu–N–C and FePc@2D-Cu–N–C. a** Calculated electrostatic potentials and the model structure of FePc@ 2D-Cu–N–C. **b** Enlarged UPS spectra of FePc@2D-Cu–N–C and 2D-FeCu–N–C. **c** Optimized structural configurations of TMB adsorbed on FePc@2D-Cu–N–C and 2D-FeCu–N–C. **d–f** Spin density diagrams (**d**) (yellow and green iso-surfaces are positive and negative spin density, respectively, isosurface = 0.004 a.u.), the corresponding magnetic moment (**e**) (μ$_{eff}$), and temperature-dependent magnetic susceptibility χ$_m$(T) and reciprocal χ$_m$ (**f**) of FePc@2D-Cu–N–C and 2D-FeCu–N–C. Inset in **f** is the corresponding electron filling in *d* orbitals, where L.S. and M.S. are low- and medium-spin, respectively. **g–h** Free-energy diagrams of the OXD-like mechanism (**g**) and the corresponding structures (**h**) of FePc@2D-Cu–N–C and 2D-FeCu–N–C surface with adsorbed intermediates. **i** Relationship between ICOHP values and the free energy of *O absorbed on different model surfaces. Source data are provided as a Source Data file.

oxygen adsorption. Remarkably, the positive spin moment of both FePc@2D-Cu–N–C and 2D-FeCu–N–C are mainly centralized on Fe sites (Fig. 5d). The locally high spin density on the Fe centers is favorable for facilitating the adsorption of oxygen and formation of oxygen intermediates. Moreover, FePc@2D-Cu–N–C possesses a higher theoretical magnetic moment (μ$_{eff}$, 2.79 μ$_B$) than 2D-FeCu–N–C (1.15 μ$_B$, Fig. 5e). To further comprehend the electron spin configuration, temperature-dependent magnetic susceptibility (χ$_m$(T)) measurements were performed. As revealed by the 1/χ$_m$ plots, the Fe species in FePc@2D-Cu–N–C display a much weaker paramagnetic state than those in 2D-FeCu–N–C, implying less Pauli paramagnetic nature of Fe centers (Fig. 5f)[59]. Concurrently, FePc@2D-Cu–N–C has a larger magnetic moment (μ$_{eff}$) value of 2.88 μ$_B$ than 2D-FeCu–N–C (0.81 μ$_B$), implying more unpaired 3*d* electrons in the FePc@2D-Cu–N–C consistent with our DFT computations[60]. This increase in the magnetic moment clearly indicates that the spin state of the central Fe cation is strongly influenced and regulated by the electronic redistribution[59].

To gain insights into the O$_2$ activation reaction, we computed free adsorption energies (ΔG) of OOH*, O*, and OH* species with corresponding reaction intermediates. The potential-determining step (PDS) is OH* → H$_2$O during O$_2$ activation reaction because of its least negative ΔG value among all elementary steps (Fig. 5g, h, Supplementary Figs. 36 and 37)[59]. Moreover, the computed ΔG value for OH* desorption on FePc@2D-Cu–N–C is −0.86 Ev closer to the equilibrium energy of this step (−1.23 eV) than that of on 2D-FeCu–N–C, highlighting the higher O$_2$ activation capacity. Considering other intermediates (such as O* + OH*) can form on single-atom catalysts and change the reaction profile drastically[61], we examined other reaction steps by considering various intermediates. As revealed by Supplementary Fig. 38, Supplementary Tables 6 to 7, the pathway of O$_2$→OOH*→O*→OH*→H$_2$O is the most favorable in the free-energy profile of FePc@2D-Cu–N–C and 2D-FeCu–N–C. For comparison, a theoretical study was performed to verify OXD-like activities of pristine FePc. The formation of OH* on the FePc is the PDS with a ΔG of

−0.75 eV, which is less negative compared to FePc@2D-Cu–N–C (−0.86 eV). These results collectively demonstrate that the 2D-Cu–N–C substrate can enhance the activity of FePc towards the OXD-like enzymatic reaction. Accordingly, the O* species anchored on the Fe center (Fe-N$_4$-O*) is identified as an essential intermediate[4], which is similar to the high-valent Fe$^{IV}$=O intermediate in the natural heme cofactors[21]. Thus, ΔG$_{O*}$ was employed to describe the OXD-like performance of these catalysts. According to the computed free-energy profiles, FePc@2D-Cu–N–C exhibits the optimal adsorption strength toward these oxygenated species with well-distributed ΔG values in each elementary step. In contrast, either too strong (such as 2D-Fe-N–C) or too weak O* adsorption (2D-Cu–N–C) leads to unsatisfied catalytic performance. Furthermore, we conducted integrated-crystal orbital Hamilton population (ICOHP) analyses to elucidate the remarkable difference in the OXD-like catalytic activity among these candidates, where the stronger binding strength is obtained by more negative ICOHP. In Fig. 5i, a strong linear scaling relationship is observed between the ICOHP values of active sites and free energies of oxygenated intermediates (*O). This correlation highlights the connection between the binding strength of active sites and energetics of the oxygenated intermediates. A similar relationship between free energies of *O and charge of the active sites is also confirmed (Supplementary Fig. 39). Importantly, the FePc@2D-Cu–N–C configuration exhibits a moderate ICOHP and charge value for the Fe active site, which can be attributed to its superior O$_2$ activation capacity.

## CYP3A4-like metabolism and DDI behaviors of two SAzymes

To explore the potential applications of FePc@2D-Cu–N–C nanozyme, which exhibits remarkable O$_2$ activation capacity, we conducted further investigations into its ability to perform O$_2$-dependent oxidative dehydrogenation for drug metabolism. 1,4-Dihydropyridine (1,4-DHP), one of the most potent calcium channel blockers, has been widely used in the treatment of cardiovascular diseases. As illustrated in Fig. 6a, the CYP3A4 enzyme, which belongs to one of the most abundant sub-families of the CYP isoforms, catalyzes the multistep metabolic

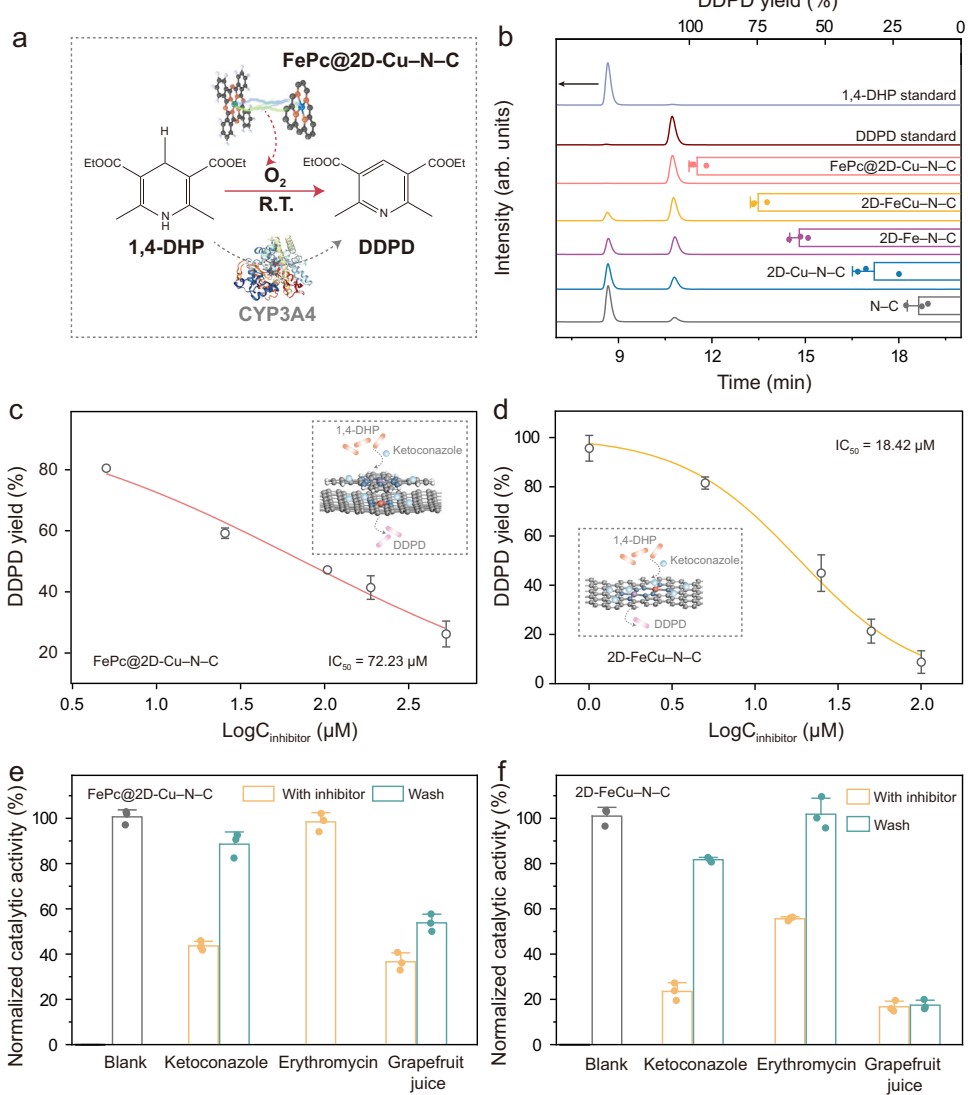

**Fig. 6 | CYP3A4-like metabolism and DDI behaviors of FePc@2D-Cu–N–C and 2D-FeCu–N–C. a** Dehydrogenation of 1,4-DHP by CYP3A4 natural enzyme and FePc@2D-Cu–N–C. **b** HPLC chromatograms of DDPD using FePc@2D-Cu–N–C as CYP3A4-mimic (*n* = 3 independent experiments). **c**, **d** Logarithmic transformation in calculating the IC$_{50}$ for the metabolization of 1,4-DHP by FePc@2D-Cu–N–C (**c**) and 2D-FeCu–N–C (**d**) (*n* = 3 independent experiments). **e**, **f** Normalized catalytic activities of FePc@2D-Cu–N–C (**e**) and 2D-FeCu–N–C (**f**) for the metabolism

of 1,4-DHP in the presence of various inhibitors: ketoconazole (FePc@2D-Cu–N–C: 100 μM; 2D-FeCu–N–C: 50 μM), erythromycin (500 μM), and grapefruit juice supernatant (a dose of 20 μL) (*n* = 3 independent experiments). Blank sample contains 0.25 mg mL$^{-1}$ of 1,4-DHP and 100 μg mL$^{-1}$ of each SAzyme. After poisoning, the inhibitor was washed with methanol three times. All data are presented as mean values ± SD. Source data are provided as a Source Data file.

processes involved in the conversion of 1,4-DHP into diethyl-2,6-dimethyl-3,5-pyridinedicarboxylate (DDPD). This metabolization of 1,4-DHP has been achieved using Fe SAzymes, which was inspired by the structural similarity between the Fe–N$_4$ active centers and the heme cofactor present in CYP. However, the potential of dual-site SAzymes in this context remains largely unexplored until now[56]. The FePc@2D-Cu–N–C dual-site SAzyme developed in this study also demonstrates a significantly higher capacity for oxidizing 1,4-DHP to DDPD compared to its single-atom Fe and Cu counterparts (Fig. 6b and Supplementary Figs. 40–42). In addition, FePc@2D-Cu–N–C displays CYP3A4-like activity comparable to natural CYP3A4 enzymes and long-term stability with over 80% retention of the initial CYP3A4-like activity even after being recycled seven times (Supplementary Fig. 43). Given the intrinsic drawbacks of homogeneous enzymatic proteins, the robust structure, excellent stability, and facile separation of FePc@2D-Cu–N–C make it a promising alternative.

The CYP3A4 is responsible for the metabolism of more than 50% of medicines. Fe SAzymes has been to exhibit enzymatic-like accelerated and inhibited behaviors akin to natural CYP, which are highly dependent on the interaction between heme-like Fe–N$_x$ coordination centers and substrates[52,62]. Our comprehensive spatial engineering strategy employed in this study enables the incorporation of single-atomic Fe active centers and Cu atomic sites (Fe–N$_4$ and Cu–N$_4$) with distinct spatial configurations. These two dual-site SAzyme models offer a more in-depth understanding of the previous studies in the field. Specifically, ketoconazole is an antifungal drug that can treat skin infections caused by a fungus and is one of the well-known CYP3A4 inhibitors. High-performance liquid chromatography (HPLC) analyses reveal that the addition of ketoconazole lowers the DDPD yield of both FePc@2D-Cu–N–C and 2D-FeCu–N–C (Supplementary Fig. 44). This inhibitory action of ketoconazole is more severe in the case of 2D-FeCu–N–C than FePc@2D-Cu–N–C. To understand the difference in

the inhibition mechanism, half-maximal inhibitory concentration ($IC_{50}$) was calculated to evaluate the informative measure of a drug's efficacy[63]. Examining the logarithmic trendlines given in Fig. 6c, d, it is evident that FePc@2D-Cu–N–C has a much larger $IC_{50}$ value (72.23 μM) than 2D-FeCu–N–C (18.42 μM), which suggests the weaker inhibitory ability of ketoconazole towards FePc@2D-Cu–N–C. Moreover, the $IC_{50}$ curve of 2D-FeCu–N–C displays a sigmoidal shape more similar to that of CYP3A4 than FePc@2D-Cu–N–C. In clinical scenarios, the binding of ketoconazole to CYP3A4 is mainly driven by the non-polar van der Waals interactions to score the shape complementarity[64]. Besides, ketoconazole typically displays a mixed reversible inhibition of CYP3A enzymes by simultaneously binding to the Fe active site in the heme[62]. Thus, the observed difference in the ketoconazole inhibition effect is likely attributed to distinct structural configurations. The single-atomic Fe active centers in 2D-FeCu–N–C adopt a planar configuration, resembling the heme cofactor in CYP. This feature results in a weak steric hindrance effect and increased likelihood of shape complementarity with ketoconazole. Conversely, FePc@2D-Cu–N–C possesses a Janus-like spatial structure with negatively charged axial Fe–$N_4$ centers and positively charged single-atom Cu layers. In the case of mixed inhibition, the substantial steric hindrance and more complex interactions with ketoconazole may lead to a weaker inhibition effect and different shape tendencies of the dose–response curves.

Erythromycin is a moderate inhibitor of the CYP3A4 system and mainly aims at benzodiazepine agents rather than the 1,4-DHP class of drugs[56]. Interestingly, erythromycin has nearly no effect on the CYP3A4-like activity of FePc@2D-Cu–N–C (Fig. 6e). The inhibition of 2D-FeCu–N–C activity by erythromycin is also much weaker than ketoconazole (Fig. 6f). On the other hand, grapefruit juice includes psoralens which induce suicide inhibition of CYP3A4 by forming a covalent bond, causing irreversible inactivation (mechanism-based inhibition) to change the pharmacokinetics of various medications[65]. As a result, grapefruit juice significantly reduces the CYP3A4-like activities of FePc@2D-Cu–N–C and 2D-FeCu–N–C, which suggests that the DDI behaviors of both CYP3A4 mimics could be different because the dissimilar spatial configurations. Due to the shared CYP450-mediated metabolic pathways, an in-depth understanding of different mechanism-based inhibition is crucial. Upon washing with methanol, FePc@2D-Cu–N–C regains the CYP3A4-like activity reduced by ketoconazole to almost its initial level. The inhibition by grapefruit juice, however, persists even after the methanol washing. Similarly, the activity of 2D-FeCu–N–C is largely recovered by washing with methanol in the case of ketoconazole and erythromycin inhibitions but remains deactivated in the case of grapefruit juice. These findings collectively suggest the feasibility of utilizing the spatial engineering strategy to regulate the CYP3A4-like biocatalytic activity and the associated inhibition mechanism (Supplementary Fig. 45).

## Discussion

In this study, we have designed and fabricated a highly optimized bio-inspired configuration of dual-site SAzymes through a comprehensive spatial engineering strategy. Employing morphological and structural investigations at the atomic level, we demonstrate that FePc@2D-Cu–N–C dual-site SAzyme, featuring vertically stacked Fe–$N_4$ and Cu–$N_4$ geometry, possesses well-integrated spatial arrangements and electronic structures compared to its single- and conventional dual-atom counterparts. Remarkably, the vertically stacked FePc@2D-Cu–N–C SAzymes exhibit significantly enhanced substrate affinity and a faster electron transfer process compared to the planar 2D-FeCu–N–C. This enables an improved biomimetic $O_2$ activation capacity and a sufficient homogenous four-electron reduction pathway, mirroring the functionality of natural CcO enzymes. Our in-depth investigations further reveal that local charge polarization and electronic coupling in the FePc@2D-Cu–N–C configuration lead to a large work function and high spin state, facilitating favorable electron

transfer from substrates. As a proof-of-concept, this proposed FePc@2D-Cu–N–C SAzyme demonstrates promising potential as a low-cost alternative to CYP in the $O_2$-dependent oxidative dehydrogenation for 1,4-DHP metabolization. Furthermore, the comprehensive spatial engineering strategy employed in this study can be applied to systematic analyses of DDI, metabolic pathways, and inhibition mechanisms. This study underscores the potential of combining heterogeneous SAzyme structures with homogeneous enzyme-like mechanisms through thorough spatial engineering, inspiring further discoveries of next-generation nanozymes.

## Methods
### Chemicals
2-Methylimidazole ($C_4H_6N_2$, 99%), iron(II) acetate ($Fe(CO_2CH_3)_2$, 95%), copper nitrate hexahydrate ($Cu(NO_3)_2 \cdot 3H_2O$, 98%), o-phenylenedia-mine (OPD, 99.5%), 3,3′,5,5′-tetramethylbenzidine (TMB, ≥99%), 2,2′-azinobis(3-ethylbenzthiazoline-6-sulfonate) (ABTS, 98%), 1,10-phe-nanthroline monohydrate ($C_{12}H_8N_2 \cdot H_2O$, 99%), graphene (few layers), dicyandiamide ($C_2H_4N_4$, 99%), melamine ($C_3H_6N_6$, 99%), aniline ($C_6H_5NH_2$, ≥99.5%), sodium borohydride ($NaBH_4$, ≥98.0%), copper(II) chloride ($CuCl_2$, 99%), copper(II) phthalocyanine ($C_{32}H_{16}CuN_8$, 99%), ketoconazole (98%), diethyl 1,4-dihydro-2,6-dimethyl-3,5-pyr-idinedicarboxylate (1,4-DHP, $C_{13}H_{19}NO_4$, >98.0%), diethyl-2,6-dime-thylpyridine-3,5-dicarboxylate (DDPD, $C_{13}H_{17}NO_4$, 99%), erythromycin ($C_{37}H_{67}NO_{13}$, 98%), acetophenone ($C_8H_8O$, 99%), and CypExpress™ 3A4 Cytochrome P450 human (CYP3A4 P450) were obtained from Sigma-Aldrich, USA. 5,5-dimethyl-1-pyrroline N-oxide (DMPO, $C_6H_{11}NO$, 98%) was supplied by Adamas-beta Chemical Co., Switzer-land. Superoxide dismutase (SOD) was obtained from Macklin Bio-chemical Co., Ltd (China). Iron(II) phthalocyanine (FePc, $C_{32}H_{16}FeN_8$, 96%) was purchased from Alfa Aesar, USA. Sodium chloride (NaCl, AR) and sodium acetate (NaAc, AR) were obtained from Sinopharm Che-mical Reagent Co., China. The commercial single-layer graphene was obtained from Nanjing XFNANO Materials Tech Co., Ltd, China. Methyl alcohol ($CH_3OH$, 99.8%), ethyl alcohol (EtOH, $C_2H_5OH$, 99.8%), acetic acid (HAc, 99.0%), dimethylsulfoxide (DMSO, 99.0%), and sulfuric acid ($H_2SO_4$, 95.0 ~ 98.0%) were bought from Duksan Pure Chemicals, Korea. Potassium thiocyanate (KSCN, AR) and dimethylformamide (DMF, >99.8%) were obtained from Acros Organics, USA. Milli-Q water was used for an aqueous solution.

### Characterization
The crystal structure of samples was analyzed using a powder XRD with Cu Kα radiation of $\lambda = 1.5418$ Å (SmartLab 9 kW Advance, Rigaku, Japan). The morphology of catalysts was recorded using field-emission SEM (SU8010, Hitachi, Japan). TEM images, selected area electron diffraction patterns, and EDS mapping were acquired by a transmis-sion electron microscope (JEM-2100F, JEOL, Japan). The HAADF-STEM images were acquired by a scanning transmission electron microscope with spherical aberration at 300 kV (FEI Themis Z, Thermo Fisher, USA) and a Cs-corrected STEM (Thermo Scientific™ Spectra 300 S/TEM) operated at 300 kV. The high-resolution low-dose iDPC-STEM images and atomic-resolution EELS analysis were collected on Cs-corrected STEM (Thermo Scientific™ Spectra 300 S/TEM) operated at 80 kV. AFM analysis was obtained by depositing an ethanolic solution of catalyst on a silicon wafer (Dimension Icon, Bruker, Germany). Raman spectra were collected using a Micro-Raman spectroscopy system (Renishaw, UK). Elemental analysis was recorded with an inductively coupled plasma optical emission spectrometer (ICP-OES, Agilent 710 Series, USA) as well as an XPS (ESCALAB-250, Thermo Fisher Scientific, USA). Fourier-transform infrared spectra (FT-IR) were acquired using the Fourier-transform spectrophotometer (Nexus 470, Thermo Fisher Scientific, USA). Zeta potential in the HAc–NaAc buffer medium (pH = 4.0) was measured using the DTS1070 Zetasizer folded capillary cells by the dynamic light scattering (DLS) equipment (Zetasizer Nano

ZS, Malvern Panalytical, UK) at room temperature (RT). For the DMF medium (pH = 7.0), the Zeta potential was measured using the universal dip cell with a DLS instrument (BeNano 180 Zeta, Bettersize Instruments Ltd., China) at RT. All suspensions were diluted to 0.002% w/w. Measurements were performed in triplicate, and the standard deviation was less than 5%. ESR spectroscopy was performed by an ESR spectrometer (EMXplus-6/1, Bruker, USA) using various spin-trapping reagents. UV–vis spectra were collected with a Cary 4000 UV–vis spectrophotometer (Agilent, USA). The element-specific X-ray absorption fine structure (XAFS) spectroscopy measurements were performed on the Fe and Co K-edges at beamline 8 C of the Pohang Light Source. The 3.0 GeV storage ring maintained a ring current of 250 mA. We used a Si(111) double crystal monochromator, and the beam intensity was decreased to 30% to eliminate higher-order harmonics. During the experiments, the slit size was 0.5 mm (vertical) × 1 mm (horizontal). The $^{57}$Fe Mössbauer spectra were obtained using an MFD-500AV (Topologic·Japan). UPS was obtained from the Thermo Nexsa G2 XPS Surface Analysis System using a He I ($hv$ = 21.22 eV) discharge lamp (ESCALAB-250, Thermo Fisher Scientific, USA). The temperature ($T$)-dependent magnetic susceptibility ($\chi$(T)) was recorded by a magnetic property measurement system (Quantum Design, USA). Effective magnetic moment ($\mu_{eff}$) and average number of the unpaired Fe 3$d$ electrons ($n$) were calculated by the following equation:

$$2.828\sqrt{\chi_m T} = \mu_{eff} = \sqrt{n(n+2)} \quad (1)$$

HPLC analysis was obtained on an Agilent 1200 series equipped with a build-in UV–vis detector and a Waters Xbridge C18 column (4.6 × 250 mm, 5 μM, USA). $^1$H and $^{13}$C nuclear magnetic resonance (NMR) spectra were gathered from an NMR spectrometer (ADVANCE III 400, Bruker, USA) at room temperature in CD$_3$OD. Electrospray ionization mass spectra (ESI-MS) were obtained from the Finni-gan MAT95 electrospray ionization mass spectrometer and the ACQUITY SQD MS system (Thermo Finnigan MAT Bremen, GER).

### Synthesis of 2D-Cu–N–C, 2D-Fe–N–C, and 2D-FeCu–N–C
In a typical synthesis for 2D-Cu–N–C, solution A was obtained by dissolving Cu(NO$_3$)$_2$·3H$_2$O (0.302 g) in absolute methanol (15 mL), followed by the addition of NaCl template (100 g) under intensive stirring at room temperature for 12 h. Separately, solution B was prepared by dispersing 2-methylimidazole (0.821 g) in absolute methanol (15 mL) under ultrasonication. Solution A and NaCl (100 g) were then poured into solution B under intensive stirring at room temperature for another 12 h. After slowly evaporating methanol in the vacuum oven at 30 °C, this obtained Cu-containing precursor was directly pyrolyzed at 750 °C for 2 h under nitrogen flow with a ramping rate of 5 °C min$^{-1}$ (150 sccm). Subsequently, this as-obtained product was immersed into 0.5 M H$_2$SO$_4$ at 60 °C for 12 h to remove NaCl templates and unstable metallic impurities. The final product was gathered by filtration and dried in a vacuum oven at 60 °C overnight. As a comparison, 2D-Fe–N–C was obtained following the same procedure by replacing Cu(NO$_3$)$_2$·3H$_2$O with Fe(CO$_2$CH$_3$)$_2$ (0.217 g). 2D-FeCu–N–C was prepared following the same procedure by replacing solution C. Specifically, solution C was prepared by first dissolving Cu(NO$_3$)$_2$·3H$_2$O (0.302 g) and Fe(CO$_2$CH$_3$)$_2$ (0.217 g) in absolute methanol (15 mL), which was subsequently added to another absolute methanolic solution (15 mL) containing 1,10-phenanthroline monohydrate (1.487 g) and NaCl (200 g) under intensive stirring at room temperature for 12 h[66].

### Synthesis of FePc@2D-Cu–N–C
The as-obtained 2D-Cu–N–C (50 mg) was dispersed in DMF (49 mL) with ultrasonication for 15 min. To this suspension, a pre-determined concentration (1 mg mL$^{-1}$, 2 mg mL$^{-1}$, and 3 mg mL$^{-1}$) of FePc (1 mL) was

added dropwise (10 μL s$^{-1}$), followed by ultrasonication for 30 min (defined as FePc-1@2D-Cu-N-C, FePc-2@2D-Cu-N-C, and FePc-3@2D-Cu-N-C). After continuous stirring at RT for 24 h, the precipitates were gathered by centrifugation (8346 × $g$) and washed with DMF and ethanol three times until the supernatant became colorless. Then, this product was washed with deionized water, collected by filtration, and freeze-dried overnight to yield final products[36]. FePc-1@2D-Cu-N-C was selected as the optimal sample.

### Evaluation of the OXD-like activity
The OXD-like activities of these as-obtained nanozymes were determined by colorimetric assay. Specifically, a series of typical TMB, OPD, and ABTS substrates were oxidized by the catalysts, and their characteristic absorption bands at different wavelengths were monitored[53]. In order to obtain optimal oxidase-like catalytic reaction conditions, optimization control experiments were conducted by adjusting the pH values (2–9), incubation temperature (10–70 °C), and incubation time (0–10 min). The relative activity was determined by the ratio of the absorbance values with the maximum value set as 100%. In a typical oxidase-like reaction, TMB (in DMSO, 25 mM, 50 μL) solution was added to HAc–NaAc (0.2 M, pH = 4.0). Subsequently, 20 μL of catalyst (100 μg mL$^{-1}$) was added and incubated at RT for 10 min. The UV–vis absorbance at 652 nm was monitored.

### Steady-state kinetic study
The kinetics assay was performed to investigate the interaction between the as-prepared catalysts and TMB substrates using the kinetic mode in a UV–vis spectrophotometer. Typically, 10 μL of nanozymes (100 μg mL$^{-1}$) and various volumes (from 0 to 300 μL) of the TMB stock solution (in DMSO, the final concentration is from 0 mM to 3 mM) were added to an oxygen-saturated HAc/NaAc buffer (0.2 M, pH = 4.0) to a final volume of 2 mL. The mixture solution was transferred into a cuvette and incubated at a 37 °C for 1 min. The absorbance value at 652 nm was immediately recorded to measure the corresponding initial rate. The apparent kinetic parameters were determined by fitting the reaction velocity values and the concentration of TMB substrates by GraphPad Prism 9 software using the Michaelis–Menten equation:

$$V = \frac{V_{max} \times [S]}{K_m + [S]} \quad (2)$$

where $V_{max}$ was the maximum reaction velocity, $V$ was the initial velocity, $K_m$ was the Michaelis–Menten constant, and [S] was the concentration of substrates.

The catalytic rate constant, $k_{cat}$, was calculated as follows:

$$k_{cat} = \frac{V_{max}}{[E]} \quad (3)$$

where [$E$] was the molar concentration of the metal atom in nanozymes, which was estimated by ICP-OES analysis.

### Specific activity calculation
Specific activity (SA) was calculated using the following formula[53]:

$$SA = \frac{V/(\varepsilon \times l) \times (\Delta A/\Delta t)}{[E]} \quad (4)$$

where $V$ was the total volume of the reaction solution (μL); $l$ was the path length of light traveling in the cuvette (cm); $\varepsilon$ was the molar absorption coefficient of substrates (TMB = 39,000 M$^{-1}$ cm$^{-1}$); A was the absorbance value (652 nm for TMB); $t$ was the OXD-like reaction time, and $\Delta A/\Delta t$ was the initial rate of change (absorbance: 652 nm min$^{-1}$).

## Detection of the reactive oxygen species by ESR spectra

The hydroxyl radicals (•OH) and superoxide ($O_2^{\cdot-}$) species were monitored by ESR spectra. In the case of •OH, the •OH trapping agent DMPO (50 mM) was added to a solution of catalyst (100 μg mL$^{-1}$) in an air-saturated HAc−NaAc buffer (pH = 4.0). The mixture was then incubated for 3 min at RT. Then the mixture was aspirated into a capillary tube for ESR measurements. For the detection of $O_2^{\cdot-}$ species, a similar procedure was taken. The catalyst (100 μg mL$^{-1}$) and DMPO (50 mM) were added to an air-saturated methanol solution. After incubating for 3 min at RT, the mixture was aspirated into a capillary tube for ESR measurements.

## DFT calculations

We performed all spin-polarized computations by means of the DFT within the Vienna ab initio simulation package (VASP.5.4.4)[67,68], in which the projector augmented wave (PAW) potential[69] was used to treat the interactions of ions with electrons. The cutoff energy was set to 550 eV to guarantee a high accuracy. The Perdew−Burke−Emzerhof (PBE) functional[70] was adopted to depict the electron exchange-correlation interactions. The convergence criteria were $10^{-5}$ eV and 0.01 eV Å$^{-1}$ for the energy and the residual force, respectively. A vacuum layer of 20 Å was employed to minimize the interactions of the adjacent periodic structures. A k-point of $5 \times 5 \times 1$ was chosen for the configuration optimization, while a denser $9 \times 9 \times 1$ grid was adopted to compute the electronic properties.

## CYP3A4-like performance of nanozyme in 1,4-DHP oxidation

For a typical test, 100 μL of 1,4-dihydropyridine (1,4-DHP, 2.5 mg mL$^{-1}$) and 20 μL of FePc@2D-Cu−N−C dispersion solution (100 μg mL$^{-1}$) were mixed in an air-saturated methanol/H$_2$O (4:1, v/v) solution to keep the volume at 1 mL. The mixed solution was sealed in a glass bottle and stirred at RT for 1 h, then centrifuged at 11,750 × $g$ for 3 min. The supernatant (200 μL) was pipetted and mixed with 10 μL of acetophenone (0.075 mg mL$^{-1}$) as internal standards. A mixture of methanol and H$_2$O (4:1, v/v) was employed for the mobile phase (flow rate: 0.5 mL min$^{-1}$). CYP3A4-like activities of the as-prepared catalysts were evaluated by the standard peak of DDPD that appeared at 10.57 min. The yields of DDPD were determined based on the following formula[56]:

$$\text{Percent yield} = \frac{\text{Actual yield}}{\text{Theoretical yield}} \times 100\% \tag{5}$$

The inhibited CYP3A4-like behavior experiments were also performed under the same processes unless otherwise specified. The curves of the logarithmic transformation in calculating the IC$_{50}$ values were processed by the GraphPad Prism 9 software. The DDPD yield without the inhibitor in the control group was taken as the 100% CYP3A4-like activity.

## Reporting summary

Further information on research design is available in the Nature Portfolio Reporting Summary linked to this article.

# Data availability

All the data supporting the findings of this study are available within the article, source data, and supplementary information files. Source data are provided as a Source Data file. This study uses publicly available data from the Protein Data Bank (PDB) under accession codes: 1OCR. Source data are provided with this paper.

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

## Acknowledgements

We acknowledge the support from the Innovation and Technology Commission and The Hong Kong Polytechnic University. L.Y.S. Lee acknowledges the support from the Research Institute for Smart Energy of the Hong Kong Polytechnic University (Q-CDA3) and the Research Grants Council of the Hong Kong SAR (PolyU15217521). K.-Y. Wong acknowledges the support from the Patrick S. C. Poon Endowed Professorship. K.-S. Lee acknowledges the funding from the National Research Foundation of Korea (NRF) grant (2021R1A2C1011415). S. C. acknowledges the startup grant from the Department of Applied Physics, the Hong Kong Polytechnic University (1-BDCM). We gratefully acknowledge the support of the University Research Facility on Chemical and Environmental Analysis (UCEA) of PolyU.

## Author contributions

Y.W. and V.-K.P. contributed equally to this work. K.-Y.W. and L.L. conceived and supervised the project. Y.W. performed the sample synthesis, characterization, and measurements. S.C. and W.W. performed the aberration-correction HAADF-STEM characterizations. K-S.L. and V.-K.P. carried out the XAS measurements and provided analysis. X.C. and G.J. performed the UPS, magnetization measurement, and analysis. J.Z. and T.Y. conducted DFT calculations. Yong Wang collected the HPLC data. Y.W. and V.-K.P. wrote the manuscript, and K.-Y.W., J.Z., S.C. and L.L. revised it. V.-K.P., thank Dr. Glatzel for your helpful comments. All the authors contributed to the whole manuscript.

## Competing interests

The authors declare no competing interests.
