## [Peer Review File · Nature Communications]

Reviewers' Comments:

Reviewer #1:

Remarks to the Author:

This manuscript reported a single-atom nanozyme that mimics the cytochrome P450. The nanozyme is composed of atomically dispersed Cu in N-doped carbon (Cu-N-C) and FePc molecule was deposited on the Cu-N-C. The authors try to correlate the oxidase-like activity to the spatial configuration of Fe-Cu, however, such structure-activity relation is not solid because the exact positioning of FePc@2D-Cu-N-C is not clear based on the characterizations of the materials. Except for the Fe/Cu spatial geometry, the oxidation states of the metal centers in the natural cytochrome c oxidase (CcO) is Fe(II) and Cu(I). But in the FePc@2D-Cu-N-C, the oxidation states differ from the CcO (Supplementary Fig. 19). The authors have conducted a thorough investigation over the enzyme-like activities and employed inhibitors for further verification. In general, I think the major weakness of the manuscript is the structure-activity relation and more specific comments are listed here:

- 1) The authors have shown the off-plane Fe-Cu pair has stronger electronic coupling compared with the in-plane Fe-Cu pair and also have shown their different oxidase-like activities. However, these are indirect evidence to support that the FePc@2D-Cu-N-C have a similar spatial geometry to CcO. In addition, the different N type in the two materials could also contribute to the different oxidase-like activity.
- 2) The fitting of EXAFS data indicates that the Fe-N4 is in the axial direction with the in-plane Cu-N4 in the FePc@2D-Cu-N-C. But the authors should compare different possible models. Very often, different models can all give good fittings. Therefore, it is not decisive evidence to show the exact coordination environment.
- 3) The EXAFS of Fe K-edge of FePc@2D-Cu-N-C resembles that of FePc. Therefore, the oxidase-like activity of FePc should be included as a control in the performance study. Meanwhile, the XAS of FePc supported on carbon nitride should be compared as well.
- 4) The measurement of Zeta potential is related to pH, solvent, and other factors. Hence, the detailed experimental conditions should be provided, so is EPR.
- 5) The TEM images in Fig.2a and 2e are the same as Supplementary Fig. 8a and 8b.
- 6) A detailed description of the mechanism illustrated in Fig. 4f should be included.
- 7) The coordination number of Fe in 2D-FeCu-N-C is 2 according to the fittings of EXAFS (Supplementary Table 2). However, this result is not consistent with the Fe/Cu ratio determined by ICP-OES. And the atomic models depicted in Fig. 3f should be modified accordingly as well.
- 8) As Cu and Fe contents are quite different (Supplementary Table 1), the equal number of active centers should be used instead of the total mass of the catalysts when comparing the OXD-like performance of different samples.
- 9) When comparing the enzyme-like activity, the activity of natural enzymes such as CYP3A4 should be provided as references.
- 10) The transition from OXD-like activity to CYP3A4-like performance is not smooth. In addition to TMB, substrates such as NADPH, NDH, glutathione, or others should be investigated to verify the specificity of the catalyst.

Reviewer #2:

Remarks to the Author:

Wang and co-workers reported the heteronuclear Fe-Cu dual-atomic centers (FePc@2D-Cu-N-C) that possess similar geometric and electronic configurations to CcO. With an axial configuration of Fe and Cu single-atomic sites, high work function, medium spin state, and moderate *O adsorption energy are realized in the FePc@2D Cu-N-C, which endows an electron transfer process similar to natural CcO, to deliver extraordinary oxidase-like performance, favorable substrate affinity, and desirable kinetics. FePc@2D-Cu-N-C SAzyme is an appealing candidate for a low-cost alternative of cytochrome P450 3A4 (CYP3A4) for drug metabolism and drug-drug interaction. However, this manuscript does not make a significant contribution in comprehensive understanding of nanozyme, and is lack of sufficient novelty. Therefore, I do not recommend the publication of this manuscript in Nature Communication as a Paper that must report very high-quality new work. To improve the quality of the manuscript, the following comments should be considered.

- (1) Novelty is very important. The authors need to emphasize the innovative points (if there is any) in the paper. In this context, the paper should be strictly compared with the previous reports. Furthermore, the idea of using FePc@2D-Cu-N-C for the drug metabolism and drug-drug interaction is exactly similar with the pioneering work (doi.org/10.1002/anie.202003949). However, no appropriate citation and comparison between the advances of those works were made, which is highly not recommended from the perspective of academic criterion.
- (2) Are the Fe atoms in FePc and the Cu atom in Cu-N-C the same single atom form? In other words, can FePc call a single atom?
- (3) Although it is hard to differentiate Fe and Cu atoms owing to their close atomic numbers, Fe atoms are brighter than Cu atoms. Thus, the descriptions of Fig. 2i should be reconsider. Moreover, all of the AC-HAADF-STEM images of FePc@2D-Cu-N-C, 2D-Cu-N-C and 2D-FeCu-N-C in Fig. 2g, Fig. 2c and Supplementary Fig. 6a exhibit single bright dot and a pair of bright dots, those results may not support the dual-atomic sites.
- (4) In Supplementary Figs. 26a to 26d, the FePc@C3N4 and FePc@Graphene are compared to FePc@2D-Cu-N-C, is it reasonable to replace the FePc@C3N4 and FePc@Graphene to FePc@Cu-C3N4 and FePc@Cu-Graphene ?
- (5) Though FePc@2D-Cu-N-C displays markedly higher OXD-like performance, the author should explain "Why the FePc@2D-Cu-N-C has markedly higher OXD-like performance" from the molecular level. For example, how the dual-atomic sites of Cu and Fe synergistic interaction between the single-atom Cu center and the axial FePc to jointly promote the OXD like activity of FePc@2D-Cu-N-C?
- (6) What role does spatial configuration of axial configuration of Fe and Cu single-atomic sites play in this OXD like activity? The structure-function relationship should be disclosed.
- (7) In the catalyst preparation part, 2D-Cu-N-C and 2D-Fe-N-C were synthesized using 2-methylimidazole as N source, however, 2D-FeCu-N-C were synthesized using 1,10-phenanthroline as N source. Since the authors compared the relative population of N species in those catalysts, N source might also contribute to configuration. Please explain the specific reason.
- (8) It is confusing to describe the FePc@2D-Cu-N-C as axial spatial configuration, it is actually parallel structure. The authors should illustrate the driving force between FePc and 2D-Cu-N-C. Moreover, what is the origin of stability?
- (9) The authors should explain the concept of "Fe Single-Atom within Cu Cofactors" since actually the 2D-Cu-N-C was used as host catalysts loading FePc.
- (10) It seems that the DFT calculations carried out in mechanism study (Figure 5h) indicating the OXD-like behavior were mainly contribute to the FePc structure. The authors should compare the activity of pure FePc with FePc@2D-Cu-N-C and it will be helpful if illustrating the contribution and connection between the two compositions.
- (11) Please check the statement in page 15 "Remarkably... making them active sites in the subsequent catalytic reaction."
- (12) In figure 3d, e, did the author use the same model to simulate different catalysts? Some figures are too small to see, for example, figure 3g.

Reviewer #3:

Remarks to the Author:

The manuscript "Atomic-Level Spatial Configuration of Fe Single-Atom within Cu Cofactors as Cytochrome P450 Nanozymes" reports a multi-technique study on a model system that mimics the structure of the an enzyme due to the presence of Fe and Cu single atoms embedded in carbon-based support. The work is relevant in the field of nanozymes and can provide useful information on the working mechanism of enzymes in catalytic reactions. The authors have performed an impressive series of experiments to prepare and characterize the systems described in the paper, and have complemented these data with theoretical calculations based on DFT. As this is my expertise, I will only comment on this part of the study.

The DFT calculations, while useful to provide additional information, suffer from several limitations. These limitations need to be pointed out in the paper (not in the supplementary!) so that the reader is alerted about the limits of validity of the reported results.

The first limitation is that these systems contain first-row transition metal atoms with magnetic ground states, but nevertheless they are treated with the inaccurate PBE functional. The results

can change considerably if one uses the more appropriate self-interaction corrected hybrid functionals (PBE0, HSE06 etc.). In a recent study Norskov et al. stated that "the blind use of GGA functionals to describe single-atom catalysts may produce inaccurate results" (Patel et al, The Journal of Physical Chemistry C 2018, 122, 29307). This is absolutely correct, in particular for atoms such as Fe and Cu.

In computing the reaction energy profile the authors consider the OOH, O, and OH intermediates, that are the common intermediates on metal electrodes. But on single-atom catalysts other intermediates can form and change drastically the reaction profile (see e.g. Barlocco et al, J Catal 2023, 417, 351.).

The reactions considered have been studied without modeling the solvent. I know this is done by several authors, but this is not a good motivation. Solvent effects can be important, and at least it should be mentioned that they must be accounted for a quantitative analysis of the reaction profile.

The authors mention the d-band center to explain the activity of the catalysts. However, there is no d-band in the complexes studied. In fact, since the Fe and Cu atoms are isolated, they give rise to discrete molecular orbitals, as in transition metal complexes. The d-band concept applies to metals, not to single atoms!

To summarize, if the authors want to include the DFT calculations in the paper, which is useful, they must also improve this part of the study. This can be done either by increasing the quality of the calculations (using a hybrid functional, include dispersion effects that are missing here, include the solvent, consider all possible intermediates and not only the most common ones, etc) or they must explicitly mention that due to the above limitations the results are purely qualitative and have no quantitative meaning.

Response to reviewer's comments:

(Hyperlink to Reviewer 1; Reviewer 2; Reviewer 3)

Response to Reviewer #1's comments

This manuscript reported a single-atom nanozyme that mimics the cytochrome P450. The nanozyme is composed of atomically dispersed Cu in N-doped carbon (Cu–N–C) and FePc molecule was deposited on the Cu–N–C.

(i) The authors try to correlate the oxidase-like activity to the spatial configuration of Fe–Cu, however, such structure-activity relation is not solid because the exact positioning of FePc@2D-Cu–N–C is not clear based on the characterizations of the materials.

(ii) Except for the Fe/Cu spatial geometry, the oxidation states of the metal centers in the natural cytochrome *c* oxidase (CcO) are Fe(II) and Cu(I). But in the FePc@2D-Cu–N–C, the oxidation states differ from the CcO (Supplementary Fig. 19).

The authors have conducted a thorough investigation over the enzyme-like activities and employed inhibitors for further verification. In general, I think the major weakness of the manuscript is the structure-activity relation.

Response: We sincerely thank the reviewer for the meticulous examination of our manuscript and for appreciating our research. We genuinely appreciate the constructive feedback provided by the reviewer, which has been helpful in enhancing the overall quality of our manuscript. The reviewer's comments have been carefully considered in the following section with comprehensive point-by-point responses.

(i) Following the reviewers comments, in the revised manuscript, we have supplied both the **morphology and structural characteristics** of the FePc@2D-Cu–N–C catalysts by carrying out additional AC HAADF-STEM characterization along with EELS analysis and iDPC-STEM observations, EXAFS fitting compared with different models, ⁵⁷Fe Mössbauer spectroscopy, and DFT calculation (see **Reviewer 1**, Question 1 and Question 2 for details). These investigation and mechanism for the structure-activity relationship are further unraveled from the perspective of experimental and theoretical studies (see **Reviewer 2**, Question 6 for details).

(ii) As reported in the literature for the catalytic cycle in the CcO oxidases (**Ref: Chem. Soc. Rev.**, **2020**, 49, 7301), two electrons are passed from two cytochrome *c*'s, through the Cu_A and cytochrome *a* site to the cytochrome *a*₃–Cu_B binuclear center, reducing the metals to the Fe²⁺ and Cu⁺ species (**Figure R1-1**). Subsequently, oxygen undergoes rapid reduction, with two electrons originating from the Fe²⁺-cytochrome *a*₃, resulting in the conversion to the ferryl oxo form (Fe⁴⁺=O).

Regarding the FePc@2D-Cu–N–C, the average coordination charges of Fe and Cu moieties are +2.75 and +1.7, respectively. Such Cu-induced synergistic effect further promotes the electron acceptance from TMB

electron donors and facilitate the reduction of Fe species. Consequently, the oxygen molecule is activated to form the Fe–N₄–O* configuration, which is similar to the high-valent Fe^{IV}=O intermediate on the FePc@2D-Cu–N–C configuration.

In summary, even though the oxidation states of the metal centers in the natural cytochrome *c* oxidase (CcO) and FePc@2D-Cu–N–C may not be identical, the electron transfer process and O₂ activation pathway exhibit a high degree of similarity in the oxidase-like catalytic cycle (see **Reviewer 1**, Question 6 for details).

Figure R1-1 General view of the catalytic cycle in the CcO enzymes together with the proposed oxidation states of the heme-copper binuclear center (BNC) cofactors (**Ref:** *Chem. Soc. Rev.*, 2020, 49, 7301).

Question 1: The authors have shown the off-plane Fe–Cu pair has stronger electronic coupling compared with the in-plane Fe–Cu pair and also have shown their different oxidase-like activities. However, these are indirect evidence to support that the FePc@2D-Cu–N–C have a similar spatial geometry to CcO. In addition, the different N type in the two materials could also contribute to the different oxidase-like activity.

Answer 1: Thank you for the valuable comments. We completely agree with the reviewer’s comment regarding the need for additional direct evidence to support the similar spatial geometry between FePc@2D-Cu–N–C and natural CcO enzymes. As suggested by the referee, we have supplied more visual characterizations using aberration-corrected high-angle annular dark-field scanning transmission electron microscopy (AC HAADF-STEM) measurements and low-temperature scanning tunneling microscopy (LT-STM). The oxidase-like activities for different N type are also further investigated by experimental and theoretical analysis. Below, we show four different characterizations that directly address the Referee’s question.

(i) AC HAADF-STEM characterization along with EELS analysis:

To provide compelling evidence, the FePc@2D-Cu–N–C catalyst was also characterized using AC HAADF-STEM in combination with atomic-resolution electron energy-loss spectroscopy (EELS) analysis at a relatively low beam current to minimize electron-beam perturbations. Our HAADF-STEM observations and EELS mapping of FePc@2D-Cu–N–C indicate that the spatial configuration of Fe and Cu species is highly correlated (**Figure R1-2**), which provides robust confirmation of the formation of Fe–Cu single-atom pairs in FePc@2D-Cu–N–C, closely resembling the spatial configuration observed in natural CcO.

(ii) iDPC-STEM characterization for the FePc@2D-Cu–N–C structures:

Considering the limitations of HAADF-STEM in accurately recognizing light elements in FePc@2D-Cu–

N–C based on Z contrast (Z, atomic number), we employed another STEM imaging mode, low-dose integrated differential phase contrast-scanning transmission electron microscopy (iDPC-STEM). This emerging technique offers higher contrast and signal-to-noise ratio compared to conventional STEM modes due to its efficient utilization of incident electrons and integration process. In addition, iDPC-STEM enables simultaneous imaging of both heavy and light elements, and its image contrast is nearly directly interpretable. The magnified iDPC-STEM image provides much enhanced signal-to-noise ratio of the planar 2D-FeCu–N–C characteristics compared with the STEM-HAADF image, consistent with the projected structure model (**Figure R1-3**). We also attempt to observe the atomic configuration of FePc@2D-Cu–N–C, as shown in **Figure R1-4**. The iDPC-STEM results reveal different stacking patterns, including single-strand graphene-like nanostructures and the pyrrole ring cavity in the phthalocyanine structure (**Ref: Adv. Funct. Mater.** **2023**, *33*, 2214062). **Figure R1-5** highlights that the detailed features of vertically stacked FePc@2D-Cu–N–C may vary in different dot pair regions depending on their orientations. Correspondingly, the possible intuitionistic schematic models are depicted. However, it is still challenging to resolve the spatial geometry of FePc@2D-Cu–N–C with the atomic resolution. Firstly, electron microscopic image is a 2D projection along the incident beam direction. The coaxial line of FePc and 2D-Cu–N–C layers may not align along the projection direction due to the overlapping and ripple of 2D-Cu–N–C substrates. Secondly, the framework of FePc@2D-Cu–N–C consists of a disordered graphitic matrix without high crystallinity and periodic structures, making it difficult to accurately identify the configuration.

(iii) LT-STM characterization for the FePc@2D-Cu–N–C structures:

While LT-STM has the capability to characterize atomic-scale structures at surfaces, obtaining atomic-level element imaging for FePc@2D-Cu–N–C catalysts is currently challenging due to technical limitations associated with surface roughness. Consequently, there are no available conditions at present to explore alternative characterization methods with direct atomic resolution. We believe, in the future, motivated by the promising results that we do have, to carry out follow-up studies to directly observe the atomic spatial structure.

(iv) Explanation of the N type effects on the oxidase-like activity

We agree with the Reviewer that the different N types in the 2D-FeCu–N–C and FePc@2D-Cu–N–C may potentially contribute to the oxidase-like activity (**Ref: Chem. Mater.** **2018**, *30*, 6431–6439; *Nat. Commun.* **2018**, *9*, 1440). In response to the Reviewer's suggestion, we first carried out a comparison of the high-resolution N 1s spectra of 2D-FeCu–N–C and FePc@2D-Cu–N–C. **Figure R1-6a** shows that the primary N type in FePc@2D-Cu–N–C is pyridinic N, whereas the majority in 2D-FeCu–N–C is graphitic N. This difference suggests the possibility of distinct local environments in two configurations. To further investigate the impact of different N types, we synthesized graphitic N-rich N–C (G-type N–C) and pyridinic N-rich N–C (P-type N–C) and evaluated their oxidase-like activities (**Figure R1-6b**). The results indicate that G-type N–C can also exhibit oxidase-like activity but with a significantly lower (< 10%) performance compared to 2D-FeCu–N–C and FePc@2D-Cu–N–C (**Figure R1-6c**). DFT calculations provide further confirmation of

the significantly enhanced O₂ activation capacity of FePc@2D-Cu-N-C compared to P-type N-C and G-type N-C (**Figure R1-6d**). Such a result reveals that the different N types contribute negligible oxidase-like performance and are not the main effect for engineering high-performance oxidase-like nanozymes.

In summary, these data further support that the Fe-Cu single-atom pairs in FePc@2D-Cu-N-C are similar to those in natural CcO. The configuration of FePc@2D-Cu-N-C differs from the 2D-FeCu-N-C. In addition, the different N type contributes neglectable OXD-like performance.

Figure R1-2 (Figure 2). (a) Theoretical FePc@2D-Cu-N-C model and (b) the structure of natural cytochrome *c* oxidase. (c) Aberration-corrected HAADF-STEM image of FePc@2D-Cu-N-C. (d) The intensity profile, atomic-resolution electron energy loss spectroscopy (EELS) elemental mapping, and (e) corresponding EELS spectra collected from the area marked with yellow rectangle in **Figure R1-1c**.

Figure R1-3. (a) HAADF-STEM and (b) STEM-iDPC images recorded on a 2D-FeCu-N-C catalyst. (c) Projected structure model of 2D-FeCu-N-C catalyst. The scale bars in (a) and (b) represent 2 Å.

Figure R1-4. (a, c) HAADF-STEM image and (b, d) STEM-iDPC image recorded on an FePc@2D-Cu-N-C catalyst.

Figure R1-5 (Figure 2). (a–e) HAADF-STEM image, (f–j) STEM-iDPC image, and (k–o) the corresponding possible schematic models on the visual plane recorded on a FePc@2D-Cu-N-C catalyst. The scale bars in (a–e) represent 2 Å.

Figure R1-6 (Figure 3a and Supplementary Fig. 28). High-resolution XPS spectra in the N 1s regions of (a) 2D-FeCu-N-C and FePc@2D-Cu-N-C catalysts and (b) G-type N-C and P-type N-C comparisons. (c) The corresponding oxidase-like activities based on the UV-vis absorption spectra of *ox*TMB. (d) Free-energy diagrams of the OXD-like mechanism for different samples.

Question 2: The fitting of EXAFS data indicates that the Fe-N₄ is in the axial direction with the in-plane Cu-N₄ in the FePc@2D-Cu-N-C. But the authors should compare different possible models. Very often, different models can all give good fittings. Therefore, it is not decisive evidence to show the exact coordination environment.

Answer 2: We appreciate the valuable suggestion from the reviewer. Indeed, different models can occasionally yield satisfactory EXAFS fittings results. We fully agree with the Reviewer's comment that EXAFS fittings are not a decisive evidence to show the exact coordination environment of FePc@2D-Cu-N-C. To address the Referee's comment, in our revised manuscript, we have supplemented our investigation with additional data on both the morphology and structural characteristics of FePc@2D-Cu-N-C catalysts by carrying out **EXAFS fitting** on different models, **DFT calculations**, **⁵⁷Fe Mössbauer spectroscopy**, and **AC HAADF-STEM characterization along with EELS analysis and iDPC-STEM technique** to elucidate the coordination environment of FePc@2D-Cu-N-C.

(i) EXAFS fitting:

In response to the referee's suggestion, we have explored two additional distinct potential models and conducted EXAFS modelling and fitting. To facilitate a coherent comparison, we have included both the experimental EXAFS data and the corresponding theoretical fits for two different structures (**Figure R1-7**). The results unambiguously demonstrate that our model is in agreement with experimental spectra.

(ii) DFT calculation:

We have also employed DFT calculations to identify the most stable configuration for the FePc@2D-Cu-N-C. To verify the interlayer spacing, the variation of the total energy of FePc@2D-Cu-N-C as a function of the interlayer distance was examined. The results show that the interlayer spacing of 3.08 Å corresponds to the most stable stacking configuration between FePc and 2D-Cu-N-C due to its lowest total energy (**Figure R1-8**). Thus, it is reasonable to choose this optimized FePc@2D-Cu-N-C configuration as illustrated by the inset in **Figure R1-8a** for subsequent analysis.

(iii) ^{57}Fe Mössbauer spectroscopy:

For further verify the coordination structure and spin polarization configuration of iron species in FePc@2D-Cu-N-C, ^{57}Fe Mössbauer spectroscopy was performed. The Mössbauer spectrum of FePc@2D-Cu-N-C in **Figure R1-9** can be accurately deconvoluted into three doublets based on their isomer shift (IS) and quadrupole splitting (QS) values (**Table R1-1**). The doublet of D3 (green) corresponds to the FePc-like square-planar Fe(II)N₄ moiety. The two prominent D2 (blue) and D1 (yellow) doublet peaks are assigned to X-FeN₄ species with a penta-coordinated rhombic mono-pyramidal structure (X = N or axial ligands) and X-FeN₄-adsorbed O₂ molecule (X-FeN₄-O₂), respectively. These results are closely associated with the vertically stacked configuration of FePc@2D-Cu-N-C and the strong interaction between the FePc centers and the 2D-Cu-N-C supports. This interaction induces enhanced O₂ adsorption and oxidase-like activity.

Figure R1-7 (Supplementary Fig. 18 a-c). a-c, Fe K-edge EXAFS fitting analysis of FePc@2D-Cu-N-C with different layer distances in R space. Note that the only the first model (a) is in good agreement with the experiment and the other models disagree with the experiment.

Figure R1-8 (Supplementary Fig. 18d). The binding energy of FePc molecules and 2D-Cu-N-C supports as a function of their layer distances.

Figure R1-9 (Figure 3f). ^{57}Fe Mössbauer spectrum and the corresponding deconvolution of FePc@2D-Cu-N-C.

Table R1-1 (Supplementary Table 4). Summary of the parameters for ^{57}Fe Mössbauer spectrum of FePc@2D-Cu-N-C.

Fe species	IS (mm s^{-1})	QS (mm s^{-1})	Content (%)
D1	0.188	1.04	51.53
D2	0.36	0.64	34.36
D3	0.178	2.65	14.11

Question 3: The EXAFS of Fe K-edge of FePc@2D-Cu-N-C resembles that of FePc. Therefore, the oxidase-like activity of FePc should be included as a control in the performance study. Meanwhile, the XAS of FePc supported on carbon nitride should be compared as well.

Answer 3: We thank the Reviewer for the valuable comments. In response to the Reviewer's suggestions, we have incorporated an examination of the oxidase-like activity of FePc as a control study in our performance analysis. Additionally, we have conducted a comparative analysis of FePc@C₃N₄ and the FePc@2D-Cu-N-C structures using their XAS spectra.

(i) The oxidase-like activity of FePc:

In the revised manuscript, we have supplied comprehensive experimental and theoretical investigations on the OXD-like behavior of pristine FePc structures. **Figure R1-10a** demonstrates that the absorbance intensity at 652 nm for FePc is lower than those of FePc@2D-Cu-N-C and 2D-FeCu-N-C. The specific activity of FePc is 26.04 U mg^{-1} , which is 2.07-folds lower than that of FePc@2D-Cu-N-C (**Figure R1-10b**). Additionally, the kinetic mechanism analysis further reveals that FePc has a weaker affinity towards TMB substrates and lower catalytic efficiency compared to FePc@2D-Cu-N-C (**Figures R1-10c, R1-11, and Table R1-2**). Moreover, a theoretical study is conducted to examine the oxidase-like catalytic activity of pristine FePc. The results indicate that the formation of OH* is the potential-determining step (PDS) with a ΔG of -0.75 eV (**Figure R1-12**), which is less negative compared to FePc@2D-Cu-N-C (-0.86 eV). These results collectively demonstrate that the 2D-Cu-N-C substrate can enhance the activity of FePc towards the

OXD-like enzymatic reaction.

(ii) The XAS of FePc supported on carbon nitride:

To further verify the critical role of the 2D-Cu-N-C support, we also anchored the FePc molecule onto a 2D carbon-based carbon nitride (C_3N_4) support ($FePc@C_3N_4$). As shown in the Fe K-edge X-ray absorption near-edge structure (XANES) diagram (**Figure R1-13**), the absorption edge position of $FePc@C_3N_4$ is similar to that of the FePc reference, suggesting that the valence state of the Fe species in $FePc@C_3N_4$ closely resemble that of pristine FePc. Moreover, the XAS spectra of $FePc@2D-Cu-N-C$ exhibit a noticeable shift towards higher energies compared to FePc and $FePc@C_3N_4$, indicating that the presence of Cu induces an overall charge transfer from Fe to the ligand orbitals. These results provide evidence that the square-planar Fe-N₄ moieties of the axial FePc are influenced by the electronic interaction with the 2D-Cu-N-C plane.

Figure R1-10 (Figure 4a–4c). (a) Absorption spectra of α -TMB and (b) specific activities (SA, $U \text{ mg}^{-1}$ metal atoms) of N-C, 2D-Cu-N-C, FePc, 2D-Fe-N-C, 2D-FeCu-N-C, and FePc@2D-Cu-N-C. (c) A spidergram comparing the parameters of the Michaelis–Menten kinetics (Michaelis constant (K_m), maximum reaction rates (V_{max}), catalytic rate constant (k_{cat}), and catalytic efficiency (k_{cat}/K_m)).

Figure R1-11 (Supplementary Fig. 32). Steady-state kinetic assays of FePc. (a) Michaelis–Menten curves with varying TMB concentration and (b) the corresponding Lineweaver–Burk plots.

Figure R1-12 (Figure 5g). Free-energy diagrams of the OXD-like mechanism.

Figure R1-13 (Supplementary Fig. 22). (a) Normalized Fe K-edge XANES spectra and (b) the Fourier transformed magnitude of the k^3 -weighted EXAFS spectra of FePc, FePc@C₃N₄, and FePc@2D-Cu-N-C.

Table R1-2 (Supplementary Table 5). Summary of the apparent Michaelis-Menten constant (K_m), maximum reaction rate (V_{max}), catalytic constant (k_{cat}), and catalytic efficiency (k_{cat}/K_m) of various catalysts as oxidase mimetic.

Catalyst	K_m (mM)	V_{max} ($\mu\text{M}\cdot\text{min}^{-1}$)	k_{cat} (s^{-1})	k_{cat}/K_m ($10^3\cdot\text{M}^{-1}\cdot\text{s}^{-1}$)
FePc	1.156	30.03	0.70	0.60
2D-Cu-N-C	1.192	24.27	1.10	0.92
2D-Fe-N-C	1.144	33.56	1.12	0.98
2D-FeCu-N-C	0.84	41.15	1.72	2.05
FePc@2D-Cu-N-C	0.604	53.48	2.43	4.02

Question 4: The measurement of Zeta potential is related to pH, solvent, and other factors. Hence, the detailed experimental conditions should be provided, so is EPR.

Answer 4: Thank you for the helpful comments and suggestions. We fully agree with the Reviewers' comments regarding the influence of pH, solvent, and temperature on the results of Zeta potential, as well as EPR. In response, we have expanded the discussion on the experimental conditions for EPR in the revised Supplementary Information as follows:

(i) **The detailed experimental conditions for Zeta potential:**

2. Characterizations

(Page S-3) Zeta potential in the HAc–NaAc buffer medium (pH = 4.0) was measured using the DTS1070 Zetasizer folded capillary cells with a dynamic light scattering (DLS) instrument (Zetasizer Nano ZS, Malvern Panalytical, UK) at room temperature. For the DMF medium (pH = 7.0), the Zeta potential was measured using the universal dip cell with a DLS instrument (BeNano 180 Zeta, Bettersize Instruments Ltd., China) at room temperature. All suspensions were diluted to 0.002% w/w. Measurements were performed in triplicate, and the standard deviation was less than 5%.

As it has been reported, zeta potential is strongly influenced by pH and salt concentration. For instance, in an acid system, the positive charges on the particle surface increase, leading to an increase in the corresponding zeta potential. As shown in **Figure R1-14**, the zeta-potential of FePc molecules shows a negative surface potential, whereas the 2D-Cu–N–C is positively charged both in the acid and neutral environment. Given the opposite charges, the negatively charged FePc molecules are expected to assemble onto the positively charged 2D-Cu–N–C through π – π stacking and electrostatic interaction.

(ii) The detailed experimental conditions for ESR spectra:

4.4. Detection of the reactive oxygen species by ESR spectra

(Page S-8 – S-9) The generation of hydroxyl radicals ($\bullet\text{OH}$) and superoxide ($\bullet\text{O}_2^-$) species were monitored by ESR spectra. In the case of $\bullet\text{OH}$, the $\bullet\text{OH}$ trapping agent DMPO (50 mM) was added to a solution of catalyst ($100 \mu\text{g mL}^{-1}$) in an air-saturated HAc–NaAc buffer (pH = 4.0). The mixture was then incubated for 3 min at room temperature. Subsequently, the mixture was aspirated into a capillary tube for ESR measurements. For the detection of $\bullet\text{O}_2^-$ species, a similar procedure was taken. The catalyst ($100 \mu\text{g mL}^{-1}$) and DMPO (50 mM) were added to an air-saturated methanol solution. After incubating for 3 min at room temperature, the mixture was aspirated into a capillary tube for ESR measurements.

Figure R1-14 (Supplementary Fig. 7). Zeta potentials of FePc, 2D-Cu–N–C, and FePc@2D-Cu–N–C in (a) the HAc–NaAc buffer medium (pH = 4.0) and (b) DMF medium (pH = 7.0).

Question 5: The TEM images in Fig.2a and 2e are the same as Supplementary Fig. 8a and 8b.

Answer 5: We thank the Referee for pointing out this careless mistake. In the revised manuscript, we have

provided the Supplementary Figure 8 correctly (**Figure R1-15**).

Figure R1-15 (Supplementary Fig. 8). TEM images of (a) FePc-2@2D-Cu-N-C and (b) FePc-3@2D-Cu-N-C.

Question 6: A detailed description of the mechanism illustrated in Fig. 4f should be included.

Answer 6: Thank you for the helpful comments.

We have reorganized the **Figure 4f** and extended the description about the mechanism in the revised Manuscript as follows:

(Page 15) “As depicted in the **Figure R1-16a**, the natural CcO can receive electrons from four cytochrome c molecules and transferring them to one oxygen molecule and four protons, resulting in the production of two water molecules. The cytochrome a_3 and Cu_B form a binuclear center that acts as the oxygen reduction site. From a kinetic perspective, nanozymes also can accept electrons from TMB substrates (acting as the electron donor) and undergo partial reduction in the initial step of the OXD-like reaction (denoted as reaction rate 1, v_1). Then, these electrons are delivered to oxygen molecules that are bound to the surface of nanozymes *via* various electron transfer pathways (denoted as reaction rate 2, v_2). As illustrated in **Figure R1-16b**, the axial FePc molecule on FePc@2D-Cu-N-C nanozymes can serve as an effective mediator for electron transfer. Due to its negative charge, the axial FePc molecule tends to non-covalently couple with positively charged TMB by electrostatic interaction and π - π stacking. The experimentally determined K_m values further confirm that axial FePc@2D-Cu-N-C exhibits a higher affinity toward TMB and higher v_1 value compared to the in-planar 2D-FeCu-N-C. With FePc@2D-Cu-N-C, the four-electron reduction of O_2 to H_2O is more likely to occur due to its high v_2 value, resulting in lower levels of toxic PROS. In contrast, the rate-limiting v_1 value on in-planar 2D-FeCu-N-C leads to insufficient reduction and slower v_2 rates. Consequently, this causes the production of multiple PROS and a slower oxygen reduction process (**Figure R1-16c**).”

Figure R1-16 (Figure 4f and Supplementary Fig. 34). Schematic diagram illustrating the OXD-like properties of (a) natural cytochrome c oxidase, (b) FePc@2D-Cu-N-C, and (c) 2D-FeCu-N-C.

Question 7: The coordination number of Fe in 2D-FeCu-N-C is 2 according to the fittings of EXAFS (Supplementary Table 2). However, this result is not consistent with the Fe/Cu ratio determined by ICP-OES. And the atomic models depicted in Fig. 3f should be modified accordingly as well.

Answer 7: We thank the referee for catching this mistake. We revisited the analysis and refitted the data by taking into account of the ICP-OES analysis of the Fe (~5.5 wt%) and Cu (~4 wt%). The revised fit results show good agreement with the experiment, and we have updated the fit results accordingly in **Figure R1-17** and **Table R1-3**.

Figure R1-17 (Figure 3d and Supplementary Fig. 17). Fe K-edge EXAFS fitting analysis of 2D-FeCu-N-C in (a) R space and (b) q space.

Table R1-3 (Supplementary Table 2). Summary of Fe K-edge EXAFS curves fitting parameters.

Sample	Path	Coordination Number, CN	Bond Length, R (Å)	Bond Disorder, $\sigma^{2[a]}$ ($\times 10^{-2}$ Å ²)	$\Delta E_0^{[b]}$ (eV)	R Factor ^[c]
2D-FeCu-N-C	Fe-N	4.0	2.02 ± 0.02	0.5 ± 0.1	4.0 ± 2.0	0.024
	Fe-C	2.0	3.14 ± 0.07	1.1 ± 0.9		
	Fe-C	2.0	3.37 ± 0.08	0.7 ± 0.4		
	Fe-Cu	1.0	4.06 ± 0.08	1.1 ± 0.9		

[a] σ^2 : Debye-Waller factor. [b] ΔE : the inner potential correction. [c] R factor: goodness of fit. *The experimental EXAFS fit

of the metal foil by fixing CN as the known crystallographic value.

Question 8: As Cu and Fe contents are quite different (Supplementary Table 1), the equal number of active centers should be used instead of the total mass of the catalysts when comparing the OXD-like performance of different samples.

Answer 8: Thank you for the helpful comments. We agree with the reviewer that the OXD-like performance of different samples should be compared with the equal number of active centers rather than the total mass of the catalysts.

As recommended in the literature titled “*Standardized assays for determining the catalytic activity and kinetics of peroxidase-like nanozymes*” (Ref: *Nature Protoc.*, 2018, 13, 1506–1520), we adopted certain definitions and equations to quantify the catalytic activity and kinetics of the nanozymes. In this context, one nanozyme activity unit (U) is defined as the amount of nanozyme that catalyzes 1 μmol of product per minute. The specific activity (SA) is defined as activity units per milligram of nanozyme. The OXD-like catalytic performance of nanozymes is also highly dependent on their kinetic constants, which relates reaction rates to substrate concentration according to the Michaelis–Menten equation. By obtaining the information on catalytic kinetics, we can quantitatively compare the OXD-like property of nanozymes in terms of the catalytic efficiency (k_{cat}/K_m) and catalytic rate constant (k_{cat}) for specific substrates. Therefore, based on the following equations, we calculated the values of the SA, k_{cat} , and k_{cat}/K_m using the molar concentration of the metal atom in the nanozymes, which is estimated by ICP-OES analysis (Figure R1-10b, Figure R1-10c, and Table R1-2).

(i) The steady-state kinetic study

The catalytic rate constant, k_{cat} , was calculated as follows:

$$k_{cat} = \frac{V_{max}}{[E]}$$

where $[E]$ is the molar concentration of the metal atom presented in the nanozymes, which is estimated by ICP-OES analysis.

(ii) The specific activity calculation

SA was calculated using the following formula:

$$SA = \frac{V/(\epsilon \times l) \times (\Delta A/\Delta t)}{[E]}$$

where V is the total volume of the reaction solution (μL), ϵ is the molar absorption coefficient of TMB ($39,000 \text{ M}^{-1} \text{ cm}^{-1}$); l is the path length of light traveling in the cuvette (cm), A is the absorbance value at 652 nm, t is the oxidase-like reaction time, and $\Delta A/\Delta t$ is the initial rate of change in absorbance at 652 nm min^{-1} .

Question 9: When comparing the enzyme-like activity, the activity of natural enzymes such as CYP3A4 should be provided as references.

Answer 9: Thank you for the helpful comments. In the revised manuscript, we have included a comparison of the metabolization performance of 1,4-DHP into DDPD catalyzed by the CYP3A4 natural enzyme, providing a basis for better comparison. As shown in **Figure R1-18a**, the engineered FePc@2D-Cu-N-C demonstrates a comparable activity to that of the natural CYP3A4 enzyme. Further more, the FePc@2D-Cu-N-C nanozyme offers the advantage of easy separation due to its heterogeneous nature. This characteristic enables its practical application and facilitates recycling. FePc@2D-Cu-N-C demonstrates excellent long-term stability with over 80% retention of the initial CYP3A4-like activity even after being recycled seven times (**Figure R1-18b**). Given the intrinsic drawbacks of homogeneous enzymatic proteins, the robust structure, excellent stability, and facile separation of FePc@2D-Cu-N-C make it a promising alternative.

Figure R1-18 (Supplementary Fig. 43). (a) HPLC chromatograms of 1,4-DHP oxidation product (DDPD) using FePc@2D-Cu-N-C and CYP3A4 natural enzyme. (b) Recovery and recyclability of FePc@2D-Cu-N-C for the oxidation of 1,4-DHP.

Question 10: The transition from OXD-like activity to CYP3A4-like performance is not smooth. In addition to TMB, substrates such as NADPH, NDH, glutathione, or others should be investigated to verify the specificity of the catalyst.

Answer 10: Thank you for the helpful comments.

(i) We have reorganized the transition from OXD-like activity to CYP3A4-like performance in the revised manuscript as follows:

(Page 18–19) “To explore the potential applications of FePc@2D-Cu-N-C nanozyme, which exhibit remarkable O₂ activation capacity, we conducted further investigations into its ability to perform O₂-dependent oxidative dehydrogenation for drug metabolism. 1,4-Dihydropyridine (1,4-DHP), one of the most potent calcium channel blockers, has been extensively used in the treatment of cardiovascular diseases. As illustrated in **Fig. 6a**, the CYP3A4 enzyme, which belongs to one of the most abundant subfamilies of the CYP isoforms, catalyzes the multistep metabolic processes involved in the conversion of 1,4-DHP into diethyl-2,6-dimethyl-3,5-pyridinedicarboxylate (DDPD). Inspired by the structural similarity between the Fe-N₄ active centers in FePc@2D-Cu-N-C and the heme cofactor present in CYP, we further evaluated the

potential application of FePc@2D-Cu-N-C in drug metabolism as a CYP3A4 mimic.”

(ii) In response to the Reviewer’s suggestion, we have examined the interfering substances, including nicotinamide adenine dinucleotide phosphate reduced (NADPH), nicotinamide adenine dinucleotide reduced (NADH), and reduced glutathione (GSH). By adopting the same evaluation procedures as those employed for TMB substrates, we observed that these interfering substances caused slight disturbance in the OXD-like reactions but without evoking distinct changes in the absorbance signal responses (**Figure R1-19**).

As reported in the literature (**Ref: J. Am. Chem. Soc.** **2020**, 142, 19602–19610), we note that the NADPH oxidase (NOX) transfers electrons from NADPH, catalyzing the reduction of molecular oxygen (O_2) via selective hydrogen abstraction and one-electron transfer from NADPH to oxygen. There is also a heme core of NOX, which plays a decisive role in the electron transfer process of NADPH substrates. Based on the remarkable O_2 activation capacity of FePc@2D-Cu-N-C, we also evaluated its potential NOX-like property. As shown in **Figure R1-20a**, the characteristic absorption peak of NADPH at 340 nm disappears when FePc@2D-Cu-N-C or 2D-FeCu-N-C mixtures are present, suggesting NADPH depletion akin to NOX-mimic activity. The slow NADPH oxidation rate observed in the N_2 saturated solution provides further evidence of the O_2 -dependent oxidative dehydrogenation of FePc@2D-Cu-N-C (**Figure R1-20b**). Additionally, FePc@2D-Cu-N-C exhibits efficient catalysis of the oxidation of NADH to NAD^+ and the oxidation of GSH to glutathione disulfide (GSSG), accompanied by the reduction of O_2 , indicating that FePc@2D-Cu-N-C can mimic the activities of natural NADH oxidase and glutathione oxidase ($GSHO_x$) enzymes (**Figures R1-21** and **R1-22**). These observations demonstrate the remarkable O_2 activation capacity of FePc@2D-Cu-N-C in enzymatic and biomimetic reactions (**Ref: Natl Sci Rev**, **2022**, 9, nwab186; *Angew. Chem. Int. Ed.* **2021**, 60, 12971).

Figure R1-19. Relative OXD-like activity of FePc@2D-Cu-N-C under the interference of NADPH, NADH, and GSH.

Figure R1-20. (a) UV-vis spectra of NADPH after treating with 2D-FeCu-N-C and FePc@2D-Cu-N-C. **(b)** Reduction of NADPH catalyzed by FePc@2D-Cu-N-C in air and N₂ atmosphere.

Figure R1-21. (a) UV-vis spectra of NADH after treating with 2D-FeCu-N-C and FePc@2D-Cu-N-C. **(b)** Reduction of NADH catalyzed by the FePc@2D-Cu-N-C in air and N₂ atmosphere.

Figure R1-22. (a) UV-vis spectra of GSH consumption after treating with 2D-FeCu-N-C and FePc@2D-Cu-N-C. **(b)** Reduction of GSH catalyzed by the FePc@2D-Cu-N-C in air and N₂ atmosphere.

Response to Reviewer #2's comments

Wang and co-workers reported the heteronuclear Fe–Cu dual-atomic centers (FePc@2D-Cu–N–C) that possess similar geometric and electronic configurations to CcO. With an axial configuration of Fe and Cu single-atomic sites, high work function, medium spin state, and moderate *O adsorption energy are realized in the FePc@2D Cu–N–C, which endows an electron transfer process similar to natural CcO, to deliver extraordinary oxidase-like performance, favorable substrate affinity, and desirable kinetics. FePc@2D-Cu–N–C SAzyme is an appealing candidate for a low-cost alternative of cytochrome P450 3A4 (CYP3A4) for drug metabolism and drug–drug interaction. However, this manuscript does not make a significant contribution in comprehensive understanding of nanozyme, and is lack of sufficient novelty. Therefore, I do not recommend the publication of this manuscript in Nature Communication as a Paper that must report very high-quality new work. To improve the quality of the manuscript, the following comments should be considered.

Response: We sincerely appreciate the Reviewer for valuable input, which helped us improve the quality of this manuscript. The Reviewer's comments were thoughtful and insightful, and we have taken them into careful consideration throughout the revision process. In response to each comment, we have provided comprehensive and detailed point-by-point responses, addressing the Reviewer's concerns and incorporating the suggestions in the following section.

Question 1: Novelty is very important. The authors need to emphasize the innovative points (if there is any) in the paper. In this context, the paper should be strictly compared with the previous reports. Furthermore, the idea of using FePc@2D-Cu–N–C for the drug metabolism and drug–drug interaction is exactly similar with the pioneering work (doi.org/10.1002/anie.202003949). However, no appropriate citation and comparison between the advances of those works were made, which is highly not recommended from the perspective of academic criterion.

Answer 1: Thank you for the important comment. In response to the Reviewer's feedback, we have revised the introduction of the manuscript to better emphasize the novelty of our work. We have also included additional experimental evidences to elucidate the morphology and structural characteristics of FePc@2D-Cu–N–C. We have also provided a mechanism analysis along with DFT calculation results to support the spatial-engineering strategy employed in our study. We believe that our work is original, as we have successfully demonstrated the following innovative points for the first time:

1. The proposed spatial-engineering strategy provides new insights into the design of nanozymes with biomimetic configurations and synergistic interaction (See **Reviewer 2**, Question 5 for details).
2. The structure-function relationship is successfully built for the biomimetic O₂ activation by spatially engineered Fe single atom sites within Cu cofactor (See **Reviewer 2**, Question 6, Question 9 for details).
3. Both heterogeneous advanced configurations and homogeneous enzyme-like mechanism similar to

natural CcO enzymes are achieved by the as-developed vertically stacked FePc@2D-Cu-N-C (See **Reviewer 2**, Question 3 and **Reviewer 1**, Question 6 for details).

We are well aware of the pioneering work of Zhang et al. in the *Angew. Chem. Int. Ed.* **2020**, *59*, 14498 (Cited as **Ref. 42** in the original manuscript and **Ref. 50** in the revised manuscript) for the drug metabolism and drug-drug interaction using the SAzymes. We have made the appropriate citation and provided a comparison between the advances made in their study and the contributions of our own work in the revised manuscript. We believe our contribution is original as we have demonstrated the followings for the first time:

(i) Different catalysts (single sites vs. dual sites)

We have employed rational design approach to construct a bioinspired spatial configuration of a Fe single-atom nanozyme incorporating Cu cofactors (FePc@2D-Cu-N-C). This design involves the incorporation for vertical stacking of Cu-N₄ moieties with axial Fe-N₄ active sites, achieving a significant enhancement in the O₂ activation activity of FePc@2D-Cu-N-C compared to its single- and dual-atom counterparts. Our investigations, combining experimental and theoretical analyses, reveal that the vertically stacked Cu-N₄ and Fe-N₄ dual-atomic active sites play a crucial role in improving the local charge polarization and electronic coupling within FePc@2D-Cu-N-C. As a result, FePc@2D-Cu-N-C exhibits a large work function and a high spin state, which facilitate favorable electron transfer from substrates. Remarkably, such an electron transfer process closely resembles the behavior observed in natural CcO enzymes. The presence of vertically stacked FePc@2D-Cu-N-C manifests several advantages over planar 2D-FeCu-N-C, particularly in terms of substrate affinity and the four-electron reduction pathway. The enhanced electron transfer capability of FePc@2D-Cu-N-C contributes to its high catalytic efficiency in the 1,4-DHP metabolism.

(ii) Different strategy and mechanism

Our work mainly focuses on the spatial-engineering strategy for constructing axial Fe single-atom sites within Cu cofactor, aiming to achieve biomimetic profiles and homogeneous enzyme-like mechanisms. In comparison to planar Fe-Cu dual-atomic centers with two-dimensional (2D) configuration (2D-FeCu-N-C), the axial tractions in vertically stacked FePc@2D-Cu-N-C confer exceptional oxidase-like performance, favorable substrate affinity, and desirable kinetics with fast electron transfer and O₂ activation process similar to natural CcO. Inspired by the structural similarity between the Fe-N₄ active centers in FePc@2D-Cu-N-C and the heme cofactor in CYP, it is reasonable to hypothesize that this bio-inspired FePc@2D-Cu-N-C nanozyme, with its remarkable O₂ activation capacity, also holds potential applications in drug metabolism as CYP3A4 mimics. More importantly, such a spatial-engineering strategy provides in-depth understanding of the intrinsic CYP3A4-like behavior and the unique inhibition mechanisms involved.

Considering these factors, we firmly believe that our work is original and distinct from previous reports in the literature.

Question 2: Are the Fe atoms in FePc and the Cu atom in Cu–N–C the same single atom form? In other words, can FePc call a single atom?

Answer 2: Thank you for the helpful comments. The local environment and coordination geometry of Fe and Cu atoms in FePc@2D-Cu–N–C have been elucidated through Fourier transform X-ray absorption fine spectroscopy (FT-EXAFS). Quantitative EXAFS fittings have provided insights into the first coordination shell of FePc@2D-Cu–N–C, which is predominantly attributed to the Fe–N scattering path with an Fe coordination number of four (N). Similarly, the local environment around Cu in FePc@2D-Cu–N–C consist of four neighboring N atoms. This observation indicates that both the Fe atom in FePc and the Cu atom in Cu–N–C possess a similar atomically dispersed metal–N₄ configuration.

We have carefully checked the related literatures for the definition of FePc as single-atom catalysts (**Ref:** *J. Am. Chem. Soc.*, **2023**, 145, 28, 15600–15610). The homogeneous FePc molecule can be defined as single-site metal atom model catalysts with Fe–N₄ active sites embedded in six carbon -membered rings. These single-site metal atoms (SMAs) on supports can be considered as the heterogenization of the molecular catalysts, which is consistent with the concept of single-atom catalysts (SACs) (**Ref:** *Acc. Chem. Res.* **2013**, 46, 8, 1740–1748; *Nat. Chem.* **2011**, 3, 634–641). This suggests that FePc supported on the 2D-Cu–N–C can be regarded as single-atom sites with well-defined Fe–N₄ moieties.

Question 3: Although it is hard to differentiate Fe and Cu atoms owing to their close atomic numbers, Fe atoms are brighter than Cu atoms. Thus, the descriptions of Fig. 2i should be reconsider. Moreover, all of the AC-HAADF-STEM images of FePc@2D-Cu–N–C, 2D-Cu–N–C and 2D-FeCu–N–C in Fig. 2g, Fig. 2c and Supplementary Fig. 6a exhibit single bright dot and a pair of bright dots, those results may not support the dual-atomic sites.

Answer 3: We thank Reviewer for the constructive comments.

Theoretically, aberration-corrected high-angle annular dark-field scanning transmission electron microscopy (HAADF-STEM) is the most commonly employed imaging mode to detect atomic sites and generate images with easy-to-interpret contrast, which is approximately proportional to Z (Z, atomic number). However, in the case of FePc@2D-Cu–N–C, it can be challenging to differentiate between Cu and Fe atoms using HAADF-STEM alone due to the low content of these species (~10 % of the total Fe or Cu species) and their isolated nature within the carbon support. We agree with the Reviewer's comment that more visual characterizations are needed to support the dual-atomic sites of FePc@2D-Cu–N–C. In response, we have tried to supply AC HAADF-STEM characterization combined with EELS analysis and the low-dose integrated differential phase contrast-scanning transmission electron microscopy (iDPC-STEM) observations.

(i) AC HAADF-STEM characterization along with EELS analysis:

The Fe and Cu atoms in the FePc@2D-Cu-N-C catalyst were further characterized using AC HAADF-STEM along with atomic-resolution electron energy-loss spectroscopy (EELS) analysis at a relatively low beam current to minimize electron-beam perturbations. Our HAADF-STEM observations and EELS mapping of FePc@2D-Cu-N-C have revealed a highly correlated spatial configuration of Fe and Cu species (**Figure R2-1**). This observation strongly support the formation of Fe-Cu dual-atom pairs within FePc@2D-Cu-N-C. In light of these results, we have made the necessary revisions to the manuscript, replacing **Figure 2i** with **Figure R2-1** to accurately reflect the findings.

(ii) iDPC-STEM characterization:

To overcome the limitations of HAADF-STEM in accurately recognizing light elements in FePc@2D-Cu-N-C, we employed the low-dose integrated differential phase contrast-scanning transmission electron microscopy (iDPC-STEM) observations. iDPC-STEM is an emerging STEM imaging mode that offers several advantages, including higher contrast and signal-to-noise ratio compared to conventional STEM modes through the efficient utilization of incident electrons and the integration process. Additionally, iDPC-STEM enables simultaneous imaging of both heavy and light elements, with image contrast that is more directly interpretable. In **Figure R2-2a**, the magnified iDPC-STEM image exhibits significantly enhanced signal-to-noise ratio of the dual-atomic site characteristics, surpassing the HAADF-STEM image. This observation is consistent with the projected 2D-FeCu-N-C model.

In our efforts to observe the atomic configuration of FePc@2D-Cu-N-C, we have noticed that the detailed features of the dual-atomic sites may vary depending on their orientations, as depicted in **Figure R2-3**. The schematic models presented provide a simplified representation of the possible atomic arrangements corresponding to different orientations and perspectives.

The electron microscopic image represents a 2D projection along the incident beam direction. However, due to the overlapping and ripple effects of FePc@2D-Cu-N-C, 2D-Cu-N-C, and 2D-FeCu-N-C, the coaxial alignment of detailed single or dual-atomic site features may not align perfectly along the projection direction. It is important to acknowledge that achieving entirely pure dual-atomic catalysts is challenging, and it is typical to find a minority of single metal sites within these catalysts (**Ref: Adv. Energy Mater.** **2021**, 2102715). To provide a more comprehensive understanding of the atomic configurations present in the catalyst, we have included **Figs. 2g** and **2c** and **Supplementary Figure 6a**, which present the results of the statistical analysis. These figures offer additional insights into the occurrence and distribution of dual single-atom motifs and single-atom sites within the FePc@2D-Cu-N-C catalyst.

Figure R2-1 (Figure 2). (a) Aberration-corrected HAADF-STEM image of the FePc@2D-Cu-N-C. (b) The intensity profile, atomic-resolution electron energy loss spectroscopy (EELS) elemental mapping, and (c) corresponding EELS spectra collected from the area marked with yellow rectangle in (a).

Figure R2-2. The (a) HAADF-STEM and (b) STEM-iDPC images recorded on a 2D-FeCu-N-C catalyst. (c) Projected structure model of 2D-FeCu-N-C catalyst. The scale bars in (a) and (b) represent 2 Å.

Figure R2-3 (Figure 2). (a–e) HAADF-STEM image, (f–j) STEM-iDPC image, and (k–o) the corresponding possible schematic models on the visual plane recorded on a FePc@2D-Cu-N-C catalyst. The scale bars in (a–e) represent 2 Å.

Question 4: In Supplementary Figs. 26a to 26d, the FePc@C₃N₄ and FePc@Graphene are compared to FePc@2D-Cu-N-C, is it reasonable to replace the FePc@C₃N₄ and FePc@Graphene to FePc@Cu-C₃N₄ and FePc@Cu-Graphene?

Answer 4: Thank you for the helpful comments and suggestions. In the revised manuscript, we have included the suggested comparisons involving C₃N₄, FePc@C₃N₄, Cu-C₃N₄, FePc@Cu-C₃N₄, graphene, Cu-graphene, FePc@graphene, and FePc@Cu-graphene (**Figure R2-4**). As displayed in **Figure R2-5**, both FePc@Cu-C₃N₄ and FePc@Cu-graphene exhibit higher OXD-like performance compared to FePc@C₃N₄ and FePc@graphene, respectively, demonstrating the advantage of the synergistic effect between Fe active sites and Cu cofactors. Furthermore, when comparing FePc@2D-Cu-N-C with FePc@Cu-C₃N₄ and FePc@Cu-graphene, there is a noticeable decrease in the OXD-like trend upon replacing 2D-Cu-N-C with other carbon-based supports. These findings provide compelling evidence for the crucial contribution of the 2D-Cu-N-C support in achieving the desired vertically stacked incorporation of axial Fe-N₄ active sites.

Figure R2-4 (Supplementary Fig. 29). TEM images of (a) FePc@C₃N₄, (b) FePc@graphene, (c) FePc@Cu-C₃N₄, and (d) FePc@Cu-graphene.

Figure R2-5 (Supplementary Fig. 30). The OXD-like performances of (a) C₃N₄, FePc, FePc@C₃N₄, Cu-C₃N₄, FePc@Cu-C₃N₄ and (b) graphene, Cu-graphene, FePc@graphene, FePc@Cu-graphene, FePc@2D-Cu-N-C.

Question 5: Though FePc@2D-Cu-N-C displays markedly higher OXD-like performance, the author should explain “Why the FePc@2D-Cu-N-C has markedly higher OXD-like performance” from the molecular level. For example, how the dual-atomic sites of Cu and Fe synergistic interaction between the single-atom Cu center and the axial FePc to jointly promote the OXD like activity of FePc@2D-Cu-N-C?

Answer 5: Thank you for the helpful suggestions. The atomic level origins of high oxidase-like catalytic behaviors of FePc@2D-Cu-N-C have been investigated through the following experimental and theoretical studies:

(i) Experimental investigation:

XANES and EXAFS analyses have provided valuable insights into the symmetry of axial FePc and its electronic interaction with the 2D-Cu-N-C plane. These analyses have revealed that the presence of the 2D-Cu-N-C support influences the symmetry of the axial FePc. Furthermore, the incorporation of more electronegative Cu elements into the interlayer is believed to effectively accelerate electron transfer and modulate the local environment of axial Fe centers within the FePc@2D-Cu-N-C framework. Such Cu-induced synergistic effects unambiguously promote the charge polarization and transfer from the axial Fe center to the single-atomic Cu sites in 2D-Cu-N-C supports. In the biomimetic O₂ activation process, the FePc@2D-Cu-N-C configuration exhibits enhanced electron transfer capacity, which contribute to more preferable electron transfer from TMB substrates to the Fe active center.

Moreover, temperature-dependent magnetic susceptibility measurements have revealed that vertically stacked FePc@2D-Cu-N-C exhibits a higher theoretical magnetic moment and more unpaired 3*d* electrons compared to planar 2D-FeCu-N-C. This observation suggests that the introduction of Cu cofactor induces the redistribution of local electron density and rearrangement of the *d*-orbital electron configuration of axial Fe centers within FePc@2D-Cu-N-C framework. The enhanced magnetic properties and increased number of unpaired electrons in FePc@2D-Cu-N-C have significant implications for oxygen adsorption and activation. These observations are reminiscent of the natural cytochrome oxidase, where the electron transfer between cytochrome *c* and the heme/Cu site is mediated by two redox cofactors, Cu_B and heme a.

(ii) Theoretical studies:

Density functional theory (DFT) calculations have provided valuable insights into the role of the 2D-Cu-N-C support in optimizing the *d*-band center of axial Fe active site within FePc@2D-Cu-N-C by exerting a traction effect. This optimization leads to the regulation of Fe-N₄-O* configuration for appropriate adsorption strength toward key intermediates, which is crucial for catalytic reactions. Through DFT calculations, it has been determined that FePc@2D-Cu-N-C exhibits the most optimal adsorption strength with well-distributed ΔG values in each elementary step, suggesting the superior OXD-like catalytic performance.

Question 6: What role does spatial configuration of axial configuration of Fe and Cu single-atomic sites play in this OXD like activity? The structure-function relationship should be disclosed.

Answer 6: We deeply appreciate this helpful suggestion. As discussed in the **Answer 5** (see **Reviewer 2, Question 5** for details), the synergistic interaction between the dual-atomic sites of Cu and Fe within FePc@2D-Cu-N-C plays a crucial role in promoting the OXD-like activity. Experimental and theoretical studies have been conducted to investigate the spatial structure-dependent O₂ activation capacity of FePc@2D-Cu-N-C. These studies provide insights into the mechanisms underlying the enhanced OXD-like features observed in the catalyst.

The inspiration drawn from the spatial architecture of heme/Cu sites in natural cytochrome *c* oxidase (CcO) offers valuable insights into the development of catalysts with enhanced O₂ activation capabilities and enhanced OXD-like features. We have carried out the design and construction of two types of oxidase-mimicking nanozymes, FePc@2D-Cu-N-C featuring axial Fe single atom sites within Cu cofactor and 2D-FeCu-N-C containing planar Fe-Cu dual-atomic centers in a two-dimensional (2D) configuration, through spatial engineering of the active sites. Experimental and theoretical analyses indicate that the vertically stacked FePc@2D-Cu-N-C exhibits stronger electronic coupling and local charge polarization compared to planar 2D-FeCu-N-C configuration, which lead to a larger work function and higher spin state, facilitating electron transfer from substrates. These advantages of FePc@2D-Cu-N-C contributes to its extraordinary oxidase-like performance, including favorable substrate affinity and desirable kinetics in the O₂ activation process. The findings suggest that the proposed spatial-engineering strategy can establish structure-function relationships for the rational design of ideal nanozymes for the biomimetic O₂ activation.

Question 7: In the catalyst preparation part, 2D-Cu-N-C and 2D-Fe-N-C were synthesized using 2-methylimidazole as N source, however, 2D-FeCu-N-C were synthesized using 1,10-phenanthroline as N source. Since the authors compared the relative population of N species in those catalysts, N source might also contribute to configuration. Please explain the specific reason.

Answer 7: Thank you for the important question. Indeed, we have also synthesized 2D-FeCu-N-C using 2-methylimidazole as N source, following the same procedure as for 2D-Cu-N-C and 2D-Fe-N-C. As shown in **Figure R2-6a**, the surface of the as-obtained sample was as smooth as 2D-Cu-N-C and 2D-Fe-N-C. However, the resulting sample showed metal aggregation, with Fe and Cu atoms forming clusters of nanoparticles on the carbon matrix (**Figure R2-6b**). This aggregation was likely due to the lack of sufficiently stable N sources. To address this issue, we replaced 2-methylimidazole with 1,10-phenanthroline as the N source based on previous report (**Ref: Nat. Commun.** 2019, 10, 4585). 1,10-phenanthroline is a well-known bidentate N-donor ligand with a rigid planar, hydrophobic, and electron-poor heteroaromatic system. These characteristics make it highly capable of forming stable complexes through strong coordination with most metal ions. As depicted in **Figure R2-7**, by using 1,10-phenanthroline as the N

source, we successfully fabricated the 2D sheet-like morphology of 2D-FeCu–N–C that are free of metal clusters or nanoparticles. Based on these results, we chose 1,10-phenanthroline as the N source for the synthesis of 2D-FeCu–N–C.

Figure R2-6. (a) TEM image and (b) XRD pattern of the 2D-FeCu–N–C prepared using 2-methylimidazole as the N source.

Figure R2-7 (Supplementary Fig. 5). (a) SEM, (b) TEM, (c) AFM images, and (d) XRD pattern of 2D-FeCu–N–C prepared using 1,10-phenanthroline as N source.

Question 8: It is confusing to describe the FePc@2D-Cu–N–C as axial spatial configuration, it is actually parallel structure. The authors should illustrate the driving force between FePc and 2D-Cu–N–C. Moreover, what is the origin of stability?

Answer 8: Thank you for the valuable comments.

(i) After carefully reviewing the relevant literature on the definition of axial coordination structures, it has been noted that the axial designing coordination is inspired by the structure of natural metal enzymes. (**Ref:** *Angew. Chem. Int. Ed.* **2021**, 60, 20803–20810; *J. Am. Chem. Soc.* **2023**, 145, 24, 13462–13468). It is understood that there may have been confusion regarding the description of the FePc@2D-Cu–N–C spatial configuration. Indeed, in the FePc@2D-Cu–N–C configuration, the Fe–N₄ and Cu–N₄ centers are introduced and anchored within the parallel interlayer frameworks, forming the FePc@2D-Cu–N–C architecture. Considering the relative positions of these two active sites, we propose that the Fe active center is vertically

stacked onto the 2D-Cu-N-C support, giving rise to the axial FePc@2D-Cu-N-C architecture, while the Fe single atom site is accompanied by the adjacent Cu-N₄ structure to form planar Fe-Cu dual-atomic centers with two-dimensional 2D-FeCu-N-C configuration. Based on the above factors, we finally employ the description of axial FePc@2D-Cu-N-C configuration and planar 2D-FeCu-N-C configuration.

(ii) The combined nanoscale and molecular approach to construct metal phthalocyanine-based hybrid materials has been successfully established, as reported in the literature (Ref: *Nature*, **2019**, 575, 639–642; *Nat. Nanotechnol.*, **2023**, 18, 160–167; *Nat. Commun.*, **2017**, 8, 14675). Building upon these strategies, the FePc@2D-Cu-N-C hybrids were prepared by supporting FePc molecules on 2D-Cu-N-C through strong non-covalent π - π stacking and electrostatic interactions, which contribute to the stability of FePc@2D-Cu-N-C and serve as the driving force between FePc and 2D-Cu-N-C. We have experimentally verified that the negatively charged FePc molecules are immobilized on the positively charged 2D-Cu-N-C, which is in agreement with the literature (Figure R2-8). Additionally, detailed morphology and structural characterization have demonstrated the good stability of FePc@2D-Cu-N-C in the OXD-like reaction (Figures R2-9 and R2-10).

Figure R2-8 (Supplementary Fig. 7). Zeta potentials of FePc, 2D-Cu-N-C, and FePc@2D-Cu-N-C (a) in the HAc-NaAc buffer medium (pH = 4.0) or (b) in the DMF medium (pH = 7.0).

Figure R2-9 (Supplementary Fig. 25). TEM image of the FePc@2D-Cu-N-C after the OXD-like reaction.

Figure R2-10 (Supplementary Fig. 26). Fe K-edge and Cu K-edge (a–b) XANES spectra and (c–d) FT-EXAFS spectra of FePc@2D-Cu–N–C before and after the OXD-like reaction.

Question 9: The authors should explain the concept of “Fe Single-Atom within Cu Cofactors” since actually the 2D-Cu–N–C was used as host catalysts loading FePc.

Answer 9: Thank you very much for the helpful comments. Indeed, when compared with C_3N_4 and graphene, the positively charged 2D-Cu–N–C serves as a more favorable support for assembling the negatively charged FePc molecules through π – π stacking and electrostatic interaction, leading to the formation of vertically stacked FePc@2D-Cu–N–C (Please see **Reviewer 2**, Question 4 for the details).

In addition to its role as an ideal host catalyst, the 2D-Cu–N–C is also crucial for facilitating synergistic interaction with axial Fe single atom sites, leading to remarkable OXD-like catalytic performance (Please see **Reviewer 2**, Question 5 for the detail). In natural biocatalysis system, cofactors are known to attach near to the substrate binding active site to facilitate substrate binding to enzyme. Their primary function is to assist in enzyme activity and participate in group-transfer reactions in metabolic processes. Drawing inspiration from these cofactors, the 2D-Cu–N–C was employed to promote charge polarization within the FePc@2D-Cu–N–C framework and accelerate electron transfer from TMB substrates to the Fe active center. The presence of the 2D-Cu–N–C induces a redistribution of local electron density and rearrangement of *d*-orbital electron configuration in the axial Fe centers. These changes enhance oxygen adsorption during the biomimetic O_2 activation process. Based on these observations, the concept of “Fe Single-Atom within Cu Cofactors” is proposed for the oxidase-mimicking nanozymes (FePc@2D-Cu–N–C).

Question 10: It seems that the DFT calculations carried out in mechanism study (Figure 5h) indicating the

OXD-like behavior were mainly contribute to the FePc structure. The authors should compare the activity of pure FePc with FePc@2D-Cu-N-C and it will be helpful if illustrating the contribution and connection between the two compositions.

Answer 10: Thank you for the helpful comments. In the revised manuscript, we have supplied the corresponding experimental and theoretical investigation for the OXD-like behavior of pure FePc structures.

The absorption spectra presented in **Figure R2-11a** demonstrate that the absorbance intensity at 652 nm for FePc is lower compared to FePc@2D-Cu-N-C and 2D-FeCu-N-C. Furthermore, the specific activity of FePc is 26.04 U/mg, which is 2.07-folds lower than that of FePc@2D-Cu-N-C (**Figure R2-11b**). The kinetic mechanism analysis reveals that FePc exhibits weaker affinity towards TMB substrates and lower catalytic efficiency compared to FePc@2D-Cu-N-C (**Figures R2-11c** and **R2-12**, and **Table R2-1**). In addition, theoretical study is conducted to investigate the oxidase-like catalytic activity of pristine FePc. The results indicate that the formation of OH* is the potential-determining step (PDS) with a ΔG value of -0.75 eV (**Figure R2-13**), which is less negative than that of FePc@2D-Cu-N-C (-0.86 eV). These findings suggest that the 2D-Cu-N-C substrate enhances the activity of FePc towards the OXD-like enzymatic reaction.

Figure R2-11 (Figure 4a–4c). (a) Absorption spectra of *ox*TMB and (b) specific activities (SA, $U \text{ mg}^{-1}$ metal atoms) of N-C, 2D-Cu-N-C, FePc, 2D-Fe-N-C, 2D-FeCu-N-C, and FePc@2D-Cu-N-C. (c) A spidergram comparing the parameters of the Michaelis–Menten kinetics (Michaelis constant (K_m), maximum reaction rates (V_{max}), catalytic rate constant (k_{cat}), and catalytic efficiency (k_{cat}/K_m)).

Figure R2-12 (Supplementary Fig. 32). Steady-state kinetic assays of FePc. (a) Michaelis–Menten curves with varying TMB concentration and (b) the corresponding Lineweaver–Burk plots.

Figure R2-13 (Figure 5g). Free-energy diagrams of the OXD-like mechanism.

Table R2-1 (Supplementary Table 5). Summary of the apparent Michaelis-Menten constant (K_m), maximum reaction rate (V_{max}), catalytic constant (k_{cat}), and catalytic efficiency (k_{cat}/K_m) of various catalysts as oxidase mimetic.

Catalyst	K_m (mM)	V_{max} ($\mu\text{M}\cdot\text{min}^{-1}$)	k_{cat} (s^{-1})	k_{cat}/K_m ($10^3\cdot\text{M}^{-1}\cdot\text{s}^{-1}$)
FePc	1.156	30.03	0.70	0.60
2D-Cu-N-C	1.192	24.27	1.10	0.92
2D-Fe-N-C	1.144	33.56	1.12	0.98
2D-FeCu-N-C	0.84	41.15	1.72	2.05
FePc@2D-Cu-N-C	0.604	53.48	2.43	4.02

Question 11: Please check the statement in page 15 “Remarkably... making them active sites in the subsequent catalytic reaction.”

Answer 11: Thank you for the helpful comments. In the revised manuscript, we have modified and extended the description for **Figure 5d** as follows:

(Page 17) “Remarkably, the positive spin moment of both FePc@2D-Cu-N-C and 2D-FeCu-N-C are mainly centralized on Fe sites. The locally high spin density on the Fe centers is favorable for facilitating the adsorption of oxygen and formation of oxygen intermediates.”

Question 12: In figure 3d, e, did the author use the same model to simulate different catalysts? Some figures are too small to see, for example, figure 3g.

Answer 12: We sincerely thank the Reviewer for pointing this out. In fact, we did not use the same model to simulate different catalysts. In the revised manuscript, we have reorganized the presentation of the corresponding models of the FePc@2D-Cu-N-C and 2D-FeCu-N-C in **Figures 3d** and **3e (Figure R2-14)**. For better reading in the revised manuscript, we have also enlarged the pre-edge part of the XANES spectra in **Figures 3g** and **3h (Figure R2-15)**.

Figure R2-14 (Figure 3d–3e). (a) Fe K-edge and (b) Cu K-edge EXAFS experimental results (open circles) and theoretical fits (solid lines) of 2D-FeCu-N-C and FePc@2D-Cu-N-C in R space.

Figure R2-15 (Figure 3g–3h). Normalized (a) Fe K-edge and (b) Cu K-edge XANES spectra of various catalysts with the corresponding reference compounds. The insets are the magnified corresponding pre-edge regions.

Response to Reviewer #3's comments

The manuscript "Atomic-Level Spatial Configuration of Fe Single-Atom within Cu Cofactors as Cytochrome P450 Nanozymes" reports a multi-technique study on a model system that mimics the structure of an enzyme due to the presence of Fe and Cu single atoms embedded in carbon-based support. The work is relevant in the field of nanozymes and can provide useful information on the working mechanism of enzymes in catalytic reactions. The authors have performed an impressive series of experiments to prepare and characterize the systems described in the paper, and have complemented these data with theoretical calculations based on DFT. As this is my expertise, I will only comment on this part of the study. To summarize, if the authors want to include the DFT calculations in the paper, which is useful, they must also improve this part of the study. This can be done either by increasing the quality of the calculations (using a hybrid functional, include dispersion effects that are missing here, include the solvent, consider all possible intermediates and not only the most common ones, etc) or they must explicitly mention that due to the above limitations the results are purely qualitative and have no quantitative meaning.

Response: We sincerely appreciate the Reviewer for helping us improve the quality of this manuscript. The Reviewer's comments were thoughtful and insightful, and we have taken them into careful consideration throughout the revision process. In response to each comment, we have provided comprehensive and detailed point-by-point responses, addressing the Reviewer's concerns and incorporating the suggestions in the following section.

Question 1: The DFT calculations, while useful to provide additional information, suffer from several limitations. These limitations need to be pointed out in the paper (not in the supplementary!) so that the reader is alerted about the limits of validity of the reported results. The first limitation is that these systems contain first-row transition metal atoms with magnetic ground states, but nevertheless they are treated with the inaccurate PBE functional. The results can change considerably if one uses the more appropriate self-interaction corrected hybrid functionals (PBE0, HSE06 etc.). In a recent study Norskov et al. stated that "the blind use of GGA functionals to describe single-atom catalysts may produce inaccurate results" (Patel et al, The Journal of Physical Chemistry C 2018, 122, 29307). This is absolutely correct, in particular for atoms such as Fe and Cu.

Response: We sincerely thank the Referee for the constructive comments.

The computations using the more reliable revised self-interaction corrected hybrid functionals (PBE0) were repeated to verify the accuracy of the initial calculations performed with the PBE functional. The detailed results, as depicted in **Figure R3-1**, demonstrate that there are only minor differences between the outcomes obtained using the PBE and PBE0 functionals, which confirms the validity of the conclusions drawn in the study.

Figure R3-1 (Supplementary Fig. 37). Free-energy diagrams of the OXD-like mechanism of FePc@2D-Cu-N-C using (a) PBE0 functionals and (b) PBE functionals.

Question 2: In computing the reaction energy profile the authors consider the OOH*, O*, and OH* intermediates, that are the common intermediates on metal electrodes. But on single-atom catalysts other intermediates can form and change drastically the reaction profile (see *e.g.*, Barlocco et al, *J. Catal.* 2023, 417, 351.).

Response: Thank you very much for the comment. We fully agree with the Reviewer’s comment that other intermediates can form and drastically change the reaction profile on single-atom catalysts (**Ref:** *J. Catal.* 2023, 417, 351–359, this paper was cited as **Ref. 56** in the revised Supplementary Information). As suggested, we have examined other reaction steps by considering various intermediates (such as O* + OH*) in the revised manuscript. As presented in **Figure R3-2**, and **Tables R3-1** and **R3-2**, the pathway of O₂ → OOH* → O* → OH* → H₂O is the most favorable in the free energy profile of FePc@2D-Cu-N-C and 2D-FeCu-N-C catalysts.

Figure R3-2 (Supplementary Fig. 39). The OXD-like reaction pathway with the corresponding structure and adsorbed intermediates of FePc@2D-Cu-N-C and 2D-FeCu-N-C.

Table R3-1 (Supplementary Table 6). The computed free energy changes (ΔG , eV) of each possible elementary step for the overall OXD-like enzymatic reaction pathway on FePc@2D-Cu-N-C. The steps marked with red represent its higher selectivity.

Elementary step	Free energy change (ΔG)
* + O ₂ + H ⁺ → *OOH	-1.50
* + O ₂ + H ⁺ → *O*OH	-0.47
*OOH + H ⁺ → *O + H ₂ O	-1.47
*O + H ⁺ → *OH	-1.09
*OH + H ⁺ → * + H ₂ O	-0.86

Table R3-2 (Supplementary Table 7). The computed free energy changes (ΔG , eV) of each possible elementary step for the overall OXD-like enzymatic reaction pathway on the 2D-FeCu-N-C. The steps marked with red represent its higher selectivity.

Elementary step	Free energy change (ΔG)
* + O ₂ + H ⁺ → *OOH	-1.28
* + O ₂ + H ⁺ → *OH*O	-0.47
* + O ₂ + H ⁺ → *O*OH	-1.51
*O*OH + H ⁺ → *O + H ₂ O	-1.91
*O*OH + H ⁺ → *OHOH	-0.59
*O + H ⁺ → *OH	-0.76
*OH + H ⁺ → * + H ₂ O	-0.74

Question 3: The reactions considered have been studied without modeling the solvent. I know this is done by several authors, but this is not a good motivation. Solvent effects can be important, and at least it should be mentioned that they must be accounted for a quantitative analysis of the reaction profile.

Response: Thank you very much for the helpful comments. We agree with the Reviewer that solvent effects can be important of the OXD-like enzymatic reaction profile. In the revised manuscript, we have thoroughly examined and assessed the impact of solvation effects using the implicit solvation model implemented in VASPsol. As displayed in **Figure R3-3**, a minimal energy difference (< 0.05 eV) is observed when considering the solvation effects. Based on the quantitative analysis conducted, it can be concluded that the

solvent effect has a negligible influence on the OXD-like enzymatic reaction.

Figure R3-3 (Supplementary Fig. 38). Free-energy diagrams of the OXD-like mechanism of FePc@2D-Cu-N-C (a) without or (b) with solvent effects.

Question 4: The authors mention the d-band center to explain the activity of the catalysts. However, there is no d-band in the complexes studied. In fact, since the Fe and Cu atoms are isolated, they give rise to discrete molecular orbitals, as in transition metal complexes. The d-band concept applies to metals, not to single atoms!

Response: Thank you for the helpful comments and suggestions. During the revision process, we have conducted integrated-crystal orbital Hamilton population (ICOHP) analyses to investigate and elucidate the remarkable difference in the OXD-like catalytic activity among these candidates. The ICOHP analyses provide insights into the binding strength, with more negative ICOHP values indicating stronger binding. In **Figure R3-4**, a strong linear scaling relationship is observed between the ICOHP values of the active sites and the free energies of oxygenated species (*O). This correlation highlights the connection between the binding strength of the active sites and the energetics of the oxygenated species. Importantly, the FePc@2D-Cu-N-C configuration exhibits a moderate ICOHP value for the Fe active site, which can be attributed to its superior OXD-like catalytic performance.

Figure R3-4 (Figure 5i). Relationship between ICOHP values and the free energy of *O absorbed on different model surfaces.

Reviewers' Comments:

Reviewer #1:

Remarks to the Author:

Overall, I think the manuscript has been improved with more structural characterizations provided, although the structure-activity relation is still not well addressed. The difficulty lies in the lack of accurate characterization methods to resolve the atomic configurations of the active sites, a general challenge in the area. Therefore, I recommend not to put too much emphasis on the structural similarity between CcO and the Fe-Cu nanozyme.

The other technical issues have been addressed in the revised manuscript. Literature citations need to be more comprehensive, such as works on diatomic nanozyme and diatomic catalyst synthesis.

Reviewer #2:

Remarks to the Author:

In this revised manuscript, I think the authors have addressed some issues from my concerns. However, there still several major concerns in novelty and data consistence.

1. As Reviewer 2, Question 1 mentioned, novelty is very important. Although the authors have demonstrated several innovative points, this work still face insufficient novelty according to the following reasons: (i) The similar proposed spatial-engineering strategy have been reported by several pioneering works (Adv. Funct. Mater. 2020, 31, 2007130; Acc. Mater. Res. 2021, 2, 534-547.). (ii) The exactly similar proposed enzyme-like mechanism regarding drug metabolism and drug-drug interaction have been also reported before (Angew. Chem. Int. Ed. 2020, 59, 14498.). Comparison between those advances have not yet carefully provided in the introduction and discussion section in the revised manuscript.
2. Some data are less reliable. As shown in Figure R1-20 and R1-21, for the NADPH oxidation, the characteristic absorption peak at 340 nm for NADPH should decreased along with the increase of the absorption peak at 260 nm for NAD⁺, however, those figure shows similar decrease trend in the substrate and the product.
3. For the drug-drug interaction experiment, although FePc@2D-Cu-N-C exhibit weaker inhibitory ability of ketoconazole than 2D-FeCu-N-C (Figure 6c, d), the IC₅₀ curve of compare sample 2D-FeCu-N-C shows a similar sigmoidal shape tendency with CYP3A4, as many dose-response curves follow this sigmoidal shape. More explanation of the specific observation is needed.
4. As Reviewer 2, Question 9 mentioned, from a perspective of biomimicking spatial configuration, this work using Cu-N-C as cofactors are quite different from the spatial structure of natural enzymes CcO or CYP3A4 using heme as cofactor (Adv. Funct. Mater. 2019, 30, 1905410.). It is hard to draw the conclusion that this work "realized the biomimetic profiles for spatial configuration." (copy from "abstract"). Also, the authors should pinpoint the definition of "cofactor", owing to the explanation provided by the authors remain a suspicion of expanding the concept.

Reviewer #3:

Remarks to the Author:

The authors have addressed the points raised in the original report. Based on the additional work performed, the exchange-correlation functional is found to be irrelevant (PBE and PBE0 provide the same results), the reaction follows the classical path and does not involve the formation of other stable intermediates, and the solvent has a negligible effect (this is certainly true if one uses an implicit model for the solvent, it may differ substantially if one uses explicit models). I am not entirely convinced by these answers as they partly contradict my personal experience, but I accept the conclusions. However, I notice that to the last point of my report the authors provided an unsatisfactory answer and did not change the paper. There is no d-band in SACs, as the orbitals of the TM atom are strongly localized (or one can say that the d-bands are very narrow). The d-band

center is a d-orbital center.

Response to Reviewer #1's comments

Comment 1: Overall, I think the manuscript has been improved with more structural characterizations provided, although the structure-activity relation is still not well addressed. The difficulty lies in the lack of accurate characterization methods to resolve the atomic configurations of the active sites, a general challenge in the area. Therefore, I recommend not to put too much emphasis on the structural similarity between CcO and the Fe–Cu nanozyme.

Response 1: We sincerely thank the reviewer for their understanding and constructive suggestions. We agree with the reviewer's suggestion to reduce the emphasis on the structural similarity between CcO and the Fe–Cu nanozyme.

In the revised manuscript, we focus on the **functional and mechanistic similarity** between CcO and the Fe–Cu nanozyme for biomimetic O₂ activation and homogeneous enzyme-like pathway rather than structural similarity. We have rephrased this perspective and extended the corresponding description accordingly (see below).

Abstract: *“(Page 2) Compared to the planar Fe–Cu dual-atomic sites with two-dimensional (2D) configuration, the vertically stacked Fe–N₄ and Cu–N₄ geometry (FePc@2D-Cu–N–C) possesses highly optimized scaffolds, favorable substrate affinity, and fast electron transfer process. These unique characteristics of FePc@2D-Cu–N–C SAzyme induces biomimetic O₂ activation through a homogenous enzymatic pathway, resembling functional and mechanistic similarity to natural CcO.”*

Introduction: *“(Page 4) Motivated by the above points, we have developed a comprehensive spatial engineering strategy to fabricate dual-site SAzymes. These SAzymes incorporate single atom Fe active centers (Fe–N₄) and Cu atomic sites (Cu–N₄) with distinct spatial configurations. Both experimental and theoretical studies indicate that the dual-site SAzyme featuring vertically stacked Fe–N₄ and Cu–N₄ geometry (FePc@2D-Cu–N–C) exhibits stronger electronic coupling and synergistic interaction compared with planar Fe–Cu pairs in two-dimensional (2D) architectures (2D-FeCu–N–C), thus enabling a similar electron transfer process to that observed in natural CcO. This unique spatial configuration of FePc@2D-Cu–N–C SAzyme leads to enhanced oxidase-like performance, surpassing the conventional 2D-FeCu–N–C and single-atomic Fe/Cu counterparts. Systematic kinetic investigations further unveil a highly favorable binding affinity to enzyme substrates and strong O₂ activation on the FePc@2D-Cu–N–C. Similar to natural CcO-like reaction pathway, the FePc@2D-Cu–N–C SAzyme facilitates four-electron reduction of O₂ to H₂O with low production of toxic PROS, achieving homogenous biomimetic O₂ activation.”*

Question 2: The other technical issues have been addressed in the revised manuscript. Literature citations need to be more comprehensive, such as works on diatomic nanozyme and diatomic catalyst synthesis.

Answer 2: We thank the Referee for this helpful suggestion.

As suggested by the Referee, we have supplied the discussion and literature citations (**Ref. 14–Ref. 18**) on diatomic nanozyme and diatomic catalyst synthesis in the revised manuscript as follows:

“(Page 3) *Recent advances have successfully produced intriguing diatomic catalysts (DACs) through various synthesis methods, including high-temperature pyrolysis, atomic layer deposition, and wet-chemistry approaches, thereby breaking the linear relationship limitation in heterogenous catalysis. Those studies have led to the discovery for diatomic iron nanozymes, Fe–Co dual-sites nanozymes, Fe–Cu hetero-binuclear nanozymes, and Mo/Zn dual SAzymes with multienzyme-mimetic biocatalysis.*”

Supplementary Literatures:

1. Hao Q, *et al.* Nickel dual-atom sites for electrochemical carbon dioxide reduction. *Nat. Synth.* **1**, 719-728 (2022).
2. Shan J, Ye C, Jiang Y, Jaroniec M, Zheng Y, Qiao S-Z. Metal-metal interactions in correlated single-atom catalysts. *Sci. Adv.* **8**, eabo0762 (2022).
3. Li B, *et al.* Diatomic iron nanozyme with lipoxidase-like activity for efficient inactivation of enveloped virus. *Nat. Commun.* **14**, 7312 (2023).
4. Zhang H, *et al.* Boosting the catalase-like activity of SAzymes via facile tuning of the distances between neighboring atoms in single-iron sites. *Angew. Chem. Int. Ed.*, e202316779 (2024).
5. Liu Y, Niu R, Deng R, Song S, Wang Y, Zhang H. Multi-enzyme co-expressed dual-atom nanozymes induce cascade immunogenic ferroptosis via activating interferon- γ and targeting arachidonic acid metabolism. *J. Am. Chem. Soc.* **145**, 8965-8978 (2023).
6. Zhang S, *et al.* Single-atom nanozymes catalytically surpassing naturally occurring enzymes as sustained stitching for brain trauma. *Nat. Commun.* **13**, 4744 (2022).
7. Ma C-B, *et al.* Guided synthesis of a Mo/Zn dual single-atom nanozyme with synergistic effect and peroxidase-like activity. *Angew. Chem. Int. Ed.* **61**, e202116170 (2022).

Response to Reviewer #2's comments

In this revised manuscript, I think the authors have addressed some issues from my concerns. However, there still several major concerns in novelty and data consistence.

Response: We express our profound gratitude for insightful suggestions, which have helped us greatly enhance the overall quality of our manuscript. We have carefully revised our manuscript with comprehensive evaluation and addressed the concerns related to novelty and data consistence throughout the revision process (further details are presented below).

Question 1: As Reviewer 2, Question 1 mentioned, novelty is very important. Although the authors have demonstrated several innovative points, this work still face insufficient novelty according to the following reasons: (i) The similar proposed spatial-engineering strategy have been reported by several pioneering works (*Adv. Funct. Mater.* 2020, 31, 2007130; *Acc. Mater. Res.* 2021, 2, 534-547). (ii) The exactly similar proposed enzyme-like mechanism regarding drug metabolism and drug–drug interaction has been also reported before (*Angew. Chem. Int. Ed.* 2020, 59, 14498). Comparison between those advances have not yet carefully provided in the introduction and discussion section in the revised manuscript.

Answer 1: We sincerely thank the Reviewer for the valuable comments.

Answer 1.1: Regarding the Reviewer's comment on a comparable spatial engineering strategy that has been proposed previously, we are cognizant of those significant research studies. In the following discussion (below), we delineate the distinctions between these existing works and our research:

(i) *Adv. Funct. Mater.* 2020, 31, 2007130

In this paper, the authors designed a nanozyme-based artificial peroxisome (*pero-nanozysome*), in which atomic Fe clusters were embedded in an atomically dispersed Fe–N₄–C matrix. This report is mainly concentrated on the structural similarity with natural enzymes. Our work extensively focused on the **functional and mechanistic similarity** between CcO and the Fe–Cu nanozyme for biomimetic O₂ activation and homogeneous enzyme-like pathway rather than structural similarity. As suggested by the Referee, the paper was cited as **Ref. 27** in the revised manuscript.

(ii) *Acc. Mater. Res.* 2021, 2, 534-547

In this review, authors focus on two major spatial-engineering strategies to design bioinspired nanozymes containing structures and compositions similar to natural enzymes: One strategy is to install specific amino acids to construct favorable spatial structures. Another is mainly focused on active sites following the principles of coordination chemistry. In our studies, the spatial engineering strategy is primarily employed to integrate heterogeneous SAzyme configurations and homogeneous enzyme-like mechanism. This paper was cited as **Ref. 31** in the revised manuscript.

Considering these factors, we firmly believe that our work is original and distinct from previous reports in the literature. We have mentioned the significance of our work which are different from other examples and extended specific descriptions in the revised manuscript as follows:

Abstract: “(Page 2) *The precise design of single-atom nanozymes (SAzymes) and understanding of their biocatalytic mechanisms hold great promise in the development of ideal substitutes for natural enzymes. While considerable efforts have been directed towards mimicking partial bio-inspired structures, the integration of heterogeneous SAzymes configurations and homogeneous enzyme-like mechanism remains a major concern and grand challenge. Herein, we employ a comprehensive spatial engineering strategy to fabricate dual-sites SAzymes with atomic Fe active center and adjacent Cu sites (Fe–N₄ and Cu–N₄). Compared to the planar Fe–Cu dual-atomic sites with two-dimensional (2D) configuration, the vertically stacked Fe–N₄ and Cu–N₄ geometry (FePc@2D-Cu–N–C) possesses highly optimized scaffolds, favorable substrate affinity, and fast electron transfer process. These unique characteristics of FePc@2D-Cu–N–C SAzyme induces biomimetic O₂ activation through a homogenous enzymatic pathway, resembling functional and mechanistic similarity to natural CcO. Furthermore, it presents an appealing alternative of cytochrome P450 3A4 (CYP3A4) for drug metabolism and feasible paradigm for systematic investigation of drug–drug interaction. These findings are expected to deepen the fundamental understanding of atomic-level design in next-generation bio-inspired nanozymes.*”

Introduction: “(Page 4) *It is worth noting that introducing axial ligand coordination as cofactors into single-atom configuration has shown promising results for engineering enzyme-like performances. However, SAzymes designed using reported spatial regulation strategies have primarily focused on mimicking partial bio-inspired structures of natural enzymes, largely overlooking the multi-spatial dimensionality and homogeneous enzymatic pathway. In essence, it is both feasible and meaningful to explore a more versatile spatial engineering strategy that can concurrently integrate heterogeneous SAzyme configurations and homogeneous enzyme-like mechanisms, thereby unlocking their full O₂ activation capacity.*”

Answer I.2: We agree with the Reviewer that the recent efforts on enzyme-like mechanism regarding drug metabolism and drug–drug interaction have been made more on the Fe SAzymes but less on the dual-sites SAzymes (*ACS Appl. Mater. Interfaces* 2018, 10, 35327–35333, *Angew. Chem. Int. Ed.* 2020, 59, 14498, cited as **Ref. 62** and **Ref. 56** in the revised manuscript). While Fe–N–C can catalyze drug metabolization and show inhibition behaviors similar to cytochrome P450 (CYP), the 1,4-DHP metabolization is a heterogeneous catalytic process associated with multiple proton-coupled electron transfer processes and reaction intermediate. The simplicity of the single-atom structure causes inevitable limitations in breaking linear scaling relationships between different adsorption and desorption species. In this regard, the introduction of another metal sites to construct intriguing dual-atomic geometry has been proposed as an extended version of SAzymes. The potential for drug metabolism and systematic investigation of drug–drug interaction of dual-site SAzymes remains largely unexplored so far. This as-developed FePc@2D-Cu–N–C dual-site SAzyme also exhibits remarkable CYP3A4-like performance, which is superior to that of single-

atom Fe and Cu nanozymes.

As pointed out by the Reviewer and references, the enzyme-like mechanism of drug–drug interaction study is mainly concentrated on the unique heme-like Fe–N_x coordination active center in Fe–N–C nanozyme, which offers such accelerated and inhibited biocatalytic activity thanks to a variety of delicate interactions with each substrate. In our as-constructed FePc@2D-Cu–N–C configuration, the main binding site for O₂ substrates is the axial Fe–N₄ active center. Therefore, similar accelerated and inhibited biocatalytic mechanism are also manifested on the FePc@2D-Cu–N–C dual-sites SAzymes. More importantly, the as-proposed comprehensive spatial engineering strategy can incorporate single atom Fe active center and Cu atomic sites (Fe–N₄ and Cu–N₄) with different spatial dimensional configuration. Those two dual-site SAzyme models can provide more in-depth understanding by contrasting different interactions with various substrates. However, the corresponding results are incremental to the previous studies in the field.

We acknowledge the pioneering work of other researchers in this field and have provided the detailed comparison between those advances with our investigation in the discussion section in the revised manuscript.

Discussion: “(Page 17-18) *This metabolization of 1,4-DHP has been achieved using Fe SAzymes, which was inspired by the structural similarity between the Fe–N₄ active centers and the heme cofactor present in CYP. However, the potential of dual-site SAzymes in this context remains largely unexplored until now. The FePc@2D-Cu–N–C dual-site SAzyme developed in this study also demonstrates a significantly higher capacity for oxidizing 1,4-DHP to DDPD compared to its single-atom Fe and Cu counterparts.*”

Discussion: “(Page 18) *Fe SAzymes have been to exhibit enzymatic-like accelerated and inhibited behaviors akin to natural CYP, which are highly dependent on the interaction between heme-like Fe–N_x coordination centers and substrates. Our comprehensive spatial engineering strategy employed in this study enables the incorporation of single-atomic Fe active centers and Cu atomic sites (Fe–N₄ and Cu–N₄) with distinct spatial configurations. These two dual-site SAzyme models offer a more in-depth understanding of the previous studies in the field.*”

Considering these factors, we firmly believe that our work is original and distinct from previous reports in the literature.

Question 2: Some data are less reliable. As shown in Figure R1-20 and R1-21, for the NADPH oxidation, the characteristic absorption peak at 340 nm for NADPH should decreased along with the increase of the absorption peak at 260 nm for NAD⁺, however, those figure shows similar decrease trend in the substrate and the product.

Answer 2: We appreciate the Reviewer’s comment on this matter. We apologize for this mistake using inappropriate substrate concentration and test conditions. During the revision process, we have reperformed

the corresponding experiments under the optimized conditions. As pointed out by the Reviewer and the relative references (**Ref:** *Natl Sci Rev*, **2022**, 9, nwab186; *J. Am. Chem. Soc.* **2020**, 142, 19602–19610), the characteristic absorption peak at 340 nm for NADH/NADPH decreased along with the increase of the absorption peak at 260 nm for NAD⁺/NADP⁺ (**Figure R2-1**). These observations demonstrate the remarkable NADH/NADPH oxidase-like capacity of FePc@2D-Cu-N-C.

Figure R2-1. UV-vis spectra of NADH (a) or NADPH (b) after treating with 2D-FeCu-N-C and FePc@2D-Cu-N-C.

Question 3: For the drug–drug interaction experiment, although FePc@2D-Cu-N-C exhibit weaker inhibitory ability of ketoconazole than 2D-FeCu-N-C (Figure 6c, d), the IC₅₀ curve of compare sample 2D-FeCu-N-C shows a similar sigmoidal shape tendency with CYP3A4, as many dose-response curves follow this sigmoidal shape. More explanation of the specific observation is needed.

Answer 3: Thanks very much for the constructive suggestion. We have reorganized the description and added more explanation of this specific observations as follows:

Discussion: “(Page 18-19) Examining the logarithmic trendlines given in **Figs. 6c and 6d**, it is evident that FePc@2D-Cu-N-C has a much larger IC₅₀ value (72.23 μM) than 2D-FeCu-N-C (18.42 μM), which suggests the weaker inhibitory ability of ketoconazole towards FePc@2D-Cu-N-C. Moreover, the IC₅₀ curve of 2D-FeCu-N-C displays a sigmoidal shape more similar to that of CYP3A4 than FePc@2D-Cu-N-C. In clinical scenarios, the binding of ketoconazole to CYP3A4 is mainly driven by the non-polar van der Waals interactions to score the shape complementarity. Besides, ketoconazole typically displays a mixed reversible inhibition of CYP3A enzymes by simultaneously binding to the Fe active site in the heme. Thus, the observed difference in the ketoconazole inhibition effect is likely attributed to distinct structural configurations. The single-atomic Fe active centers in 2D-FeCu-N-C adopt a planar configuration, resembling the heme cofactor in CYP. This feature results in a weak steric hindrance effect and increased likelihood of shape complementarity with ketoconazole. Conversely, FePc@2D-Cu-N-C possesses a Janus-like spatial structure with negatively charged axial Fe-N₄ centers and positively charged single-atom Cu layers. In the case of mixed inhibition, the substantial steric hindrance and more complex interactions with ketoconazole may lead to a weaker inhibition effect and different shape tendencies of the dose–response curves.”

Question 4: As Reviewer 2, Question 9 mentioned, from a perspective of biomimicking spatial configuration, this work using Cu–N–C as cofactors is quite different from the spatial structure of natural enzymes CcO or CYP3A4 using heme as cofactor (*Adv. Funct. Mater.* 2019, 30, 1905410). It is hard to draw the conclusion that this work “realized the biomimetic profiles for spatial configuration.” (copy from “abstract”). Also, the authors should pinpoint the definition of “cofactor”, owing to the explanation provided by the authors remain a suspicion of expanding the concept.

Answer 4: We are grateful for the insightful comments and suggestions, and apologize for the confusion.

We agree with the reviewer that the definition of “cofactor” for 2D-Cu–N–C is probably one-sided lacking of comprehensive consideration. The definition of “cofactor” in the biochemistry field is a non-protein chemical compound or metallic ion that is required for an enzyme's role as a catalyst (attach near to the substrate binding active site to facilitate substrate binding to enzyme). If the cofactor is removed from a complete enzyme (holoenzyme), the protein component (apoenzyme) no longer has catalytic activity. As provided in the reference (*Adv. Funct. Mater.* 2019, 30, 1905410, cited as **Ref. 10** in the revised manuscript), mimicking the active site structures of the heme cofactor in natural peroxidase enzyme is the main point. The Fe–N₄ structure is also the indispensable active site for the intrinsic peroxidase-like catalytic features.

As for this work, the main roles of 2D-Cu–N–C in the whole FePc@2D-Cu–N–C framework are to enhance oxygen adsorption on the axial Fe active sites and facilitate biomimetic O₂ activation. The main binding site for O₂ substrates in the FePc@2D-Cu–N–C is the axial Fe–N₄ active center. 2D-Cu–N–C is incorporated with vertically stacked single-atomic Fe active centers to construct multi-spatial dimensionality and thereby improve homogeneous enzyme-like mechanism. Therefore, it is more reasonable to consider the 2D-Cu–N–C as “helper molecules” than “cofactor”. We have revised the corresponding definition and discussion about “cofactor” in the revised manuscript.

In the perspective of biomimicking spatial configuration, the axial single atom Fe–N₄ active center in the FePc@2D-Cu–N–C configuration is also similar to the active site structure of the heme cofactor. The adsorption and activation processes of oxygen substrates proceed dominantly on the Fe–N₄ active sites. Moreover, the cytochrome a₃ and Cu_B in the natural CcO enzymes form a binuclear center as the oxygen-binding site for the four-electron reduction of oxygen to water. From a perspective of biomimicking spatial configuration, we fabricated dual-site SAzymes featuring vertically stacked Fe–N₄ and Cu–N₄ geometry (FePc@2D-Cu–N–C) to mimic the 3D spatial iron–copper binuclear topologies in natural CcO enzyme. Additionally, we aimed more to integrate heterogeneous SAzyme configurations and homogeneous enzyme-like mechanism. Thus, in this work, more attention is given to the biomimetic profiles towards functional and mechanistic similarity to natural CcO enzymes by a comprehensive spatial engineering strategy. To clarify the above discussion, we have reorganized the abstract, introduction, discussion, and conclusion parts in the revised manuscript.

Response to Reviewer #3's comments

Question 1: The authors have addressed the points raised in the original report. Based on the additional work performed, the exchange-correlation functional is found to be irrelevant (PBE and PBE0 provide the same results), the reaction follows the classical path and does not involve the formation of other stable intermediates, and the solvent has a negligible effect (this is certainly true if one uses an implicit model for the solvent, it may differ substantially if one uses explicit models). I am not entirely convinced by these answers as they partly contradict my personal experience, but I accept the conclusions.

Response 1: We sincerely appreciate the reviewer's meticulous examination of our manuscript. We also thank the reviewer for providing insightful expert comments. As pointed by the Reviewers, we further examined the catalytic performance by means of the revised Perdew-Burke-Emzerhof (rPBE) to verify our calculation results, which has been widely employed in electrocatalysis (**Figure R3-1**) (Ref: *J. Mater. Chem. A* 2020, 8, 17078; *Nat. Catal.* 2022, 5, 109–118). Furthermore, the solvent effect was also taken into consideration by employing both the implicit and explicit solvation models (**Figure R3-2**). Our result shows that there is a small difference between the results of PBE and rPBE, or using implicit and explicit model for the solvent, further validating the enhanced catalytic performance of FePc@2D-Cu-N-C.

Figure R3-1 (Supplementary Fig. 36). Free-energy diagrams of the OXD-like mechanism of FePc@2D-Cu-N-C using PBE0 functionals (a), PBE functionals (b), and rPBE functionals (c).

Figure R3-2 (Supplementary Fig. 37). Free-energy diagrams of the OXD-like mechanism of FePc@2D-Cu-N-C without

solvent effects (a) with implicit solvent effects (b), with explicit solvent effects (c), and the corresponding models with solvent effects (d).

Question 2: However, I notice that to the last point of my report the authors provided an unsatisfactory answer and did not change the paper. There is no d-band in SACs, as the orbitals of the TM atom are strongly localized (or one can say that the *d*-bands are very narrow). The d-band center is a *d*-orbital center.

Response 2: We appreciate the Reviewer's comment. In the revised manuscript, we have removed the discussion on the variation of catalytic trend with the *e*-band center. To address the Referee's concern, we have examined the relationship of catalytic performance reflected by the *O binding strength with the carrying charge of activity site. Remarkably, there is also a strong linear scaling relationship between each other as shown in **Figure R3-3**, from which Fe active site within the FePc@2D-Cu-N-C structure has a moderate charge among all catalysts, further rationalizing its high OXD-like catalytic activity.

Figure R3-3 (Supplementary Fig. 39). Relationship between charges of active sites and the free energy of *O adsorbed on different model surfaces.

Reviewers' Comments:

Reviewer #1:

Remarks to the Author:

All my questions have been addressed and the manuscript is in good shape for publication.

Reviewer #2:

Remarks to the Author:

In this revised version, most comments of the referees have been well addressed, and the quality of the manuscript has been substantially improved. Thus, I suggest the acceptance of this nice contribution.

Reviewer #3:

Remarks to the Author:

The authors have performed additional calculations and analyses to reinforce their conclusions. The effort is appreciated and the paper can now be recommended for publication.